# Cohesin is required for meiotic spindle assembly independent of its role in cohesion in *C. elegans*

**Karen P. McNally**[1], **Elizabeth A. Beath**[1], **Brennan M. Danlasky**[1], **Consuelo Barroso**[2], **Ting Gong**[1], **Wenzhe Li**[1], **Enrique Martinez-Perez**[2,3], **Francis J. McNally**[1]*

**1** Department of Molecular and Cellular Biology University of California, Davis, California, United States of America, **2** MRC London Institute of Medical Sciences London, United Kingdom, **3** Imperial College Faculty of Medicine London, United Kingdom

* fjmcnally@ucdavis.edu

**Data Availability Statement:** All relevant data are within the paper and its Supporting information files.

## Abstract

Accurate chromosome segregation requires a cohesin-mediated physical attachment between chromosomes that are to be segregated apart, and a bipolar spindle with microtubule plus ends emanating from exactly two poles toward the paired chromosomes. We asked whether the striking bipolar structure of *C. elegans* meiotic chromosomes is required for bipolarity of acentriolar female meiotic spindles by time-lapse imaging of mutants that lack cohesion between chromosomes. Both a *spo-11 rec-8 coh-4 coh-3* quadruple mutant and a *spo-11 rec-8* double mutant entered M phase with separated sister chromatids lacking any cohesion. However, the quadruple mutant formed an apolar spindle whereas the double mutant formed a bipolar spindle that segregated chromatids into two roughly equal masses. Residual non-cohesive COH-3/4-dependent cohesin on separated sister chromatids of the double mutant was sufficient to recruit haspin-dependent Aurora B kinase, which mediated bipolar spindle assembly in the apparent absence of chromosomal bipolarity. We hypothesized that cohesin-dependent Aurora B might activate or inhibit spindle assembly factors in a manner that would affect their localization on chromosomes and found that the chromosomal localization patterns of KLP-7 and CLS-2 correlated with Aurora B loading on chromosomes. These results demonstrate that cohesin is essential for spindle assembly and chromosome segregation independent of its role in sister chromatid cohesion.

## Author summary

Meiosis is the process that reduces the number of chromosomes from four to one during the formation of eggs and sperm so that a fertilized egg has exactly two copies of each chromosome. Meiotic errors result in offspring with an incorrect number of chromosomes which results in prenatal death or birth defects. Accurate meiosis requires that the four chromosomes at the beginning of meiosis are attached to each other by a protein called cohesin and a structure called a spindle that pulls individual chromosomes in two directions. Here we show that in the roundworm, *C. elegans*, cohesin is required for

**Funding:** This work was funded by National Institute of General Medical Sciences grant R35GM136241 to F.J.M, U.S. Department of Agriculture/National Institute of Food and Agriculture Hatch project 1009162 to F.J.M. and Medical Research Council core-funded grant MC-A652-5PY60 to E.M.P. The funders had no role in study design, data collection and analysis, decision to publish, or preparation of the manuscript.

**Competing interests:** The authors have declared that no competing interests exist.

building a spindle that can pull in two directions independently of its role in attaching chromosome copies to each other. Because cohesin is gradually lost in aging women, these results may clarify why aging women have an increasing incidence of babies with birth defects caused by an incorrect number of chromosomes.

## Introduction

The accurate segregation of chromosomes during meiosis and mitosis requires sister chromatid cohesion (SCC) provided by the cohesin complex and a bipolar spindle with microtubule minus ends oriented toward the two poles and microtubule plus ends extending from the two poles toward the chromosomes [1]. During mitosis in most animal cells, spindle formation is initiated when organelles known as centrosomes are duplicated and move to opposite sides of the cell. There they anchor, nucleate and stabilize microtubules with their plus ends polymerizing away from the poles [2]. Microtubule plus ends puncture the nuclear membrane and capture the kinetochores of chromosomes, thus establishing a symmetric spindle axis.

In contrast to the pathway of mitotic spindle formation, the female meiotic cells of many animals lack centrosomes and spindle formation initiates when microtubules organize around chromatin during the two consecutive meiotic divisions. In *Xenopus* egg extracts and mouse oocytes, DNA-coated beads are sufficient to induce bipolar spindle assembly [3,4]. The mechanisms of acentrosomal spindle assembly are being elucidated in several species and two alternate pathways have been implicated. The first molecular activity to be identified in the assembly of microtubules around meiotic chromatin is the GTPase Ran. In the Ran pathway, spindle assembly factors (SAFs) contain nuclear localization sequences and are imported into the nucleus during interphase by binding to importins. GTP-Ran, which is maintained at a high concentration in the nucleus by the chromatin-bound GEF RCC1, causes dissociation of the SAFs from importins inside the nucleus, thus driving the directionality of import. Upon nuclear envelope breakdown, tubulin enters the region adjacent to chromatin and the locally activated SAFs initiate MT nucleation and stabilization [5]. Inhibition of the Ran pathway prevents or affects the assembly of acentrosomal spindles in *Xenopus* egg extracts [6] and in mouse [7], *Drosophila* [8] and *C. elegans* oocytes [9]. In *Xenopus* egg extracts, spindle assembly is induced by beads coated with the Ran GEF, RCC1, even without DNA [10].

The second pathway which has been implicated in acentrosomal spindle assembly requires the Chromosomal Passenger Complex (CPC), which includes the chromatin-targeting proteins Survivin and Borealin, the scaffold subunit INCENP, and Aurora B kinase [11]. The CPC is recruited to distinct regions on mitotic chromosomes by at least three different pathways [12]. Depletion of CPC components resulted in a lack of spindle microtubules in *Drosophila* oocytes [13] and in *Xenopus* egg extracts to which sperm nuclei or DNA-coated beads are added [14–16]. In *C. elegans* oocytes, the CPC subunits, BIR-1/survivin [17], INCENP [18], and the Aurora B-homolog AIR-2 [19,20] contribute to meiotic spindle assembly.

While the GTP Ran and CPC pathways are known to be involved in the initiation of acentrosomal spindle assembly, the mechanism by which the microtubules are captured into two poles is unclear. Spindles with one or more poles form when chromatin-coated beads are added to *Xenopus* egg extracts, suggesting that pole formation is an intrinsic activity of microtubules assembling around chromatin [10]. However, the results also suggest that the reproducible production of bipolar spindles requires that the process includes some bidirectionality. In *C. elegans*, meiotic bivalents, which promote assembly of a bipolar metaphase I spindle, are composed of 4 chromatids held together by chiasmata, physical attachments provided by

cohesin and a single crossover formed between homologous chromosomes. These bivalents have a discrete bipolar symmetry with a mid-bivalent ring containing the CPC, and they are capped at their two ends by cup-shaped kinetochores. Metaphase II univalents, which promote assembly of a bipolar metaphase II spindle, are composed of 2 chromatids held together by cohesin. These univalents also have a discrete bipolar symmetry with a CPC ring between sister chromatids that are each capped by cup-shaped kinetochores [18,19,21].

To test whether this chromosomal bipolar symmetry is required for spindle bipolarity, we analyzed cohesin mutants that start meiotic spindle assembly with separated sister chromatids rather than the bivalents present in wild-type meiosis I or the univalents present in wild-type meiosis II. During meiosis, cohesin is composed of SMC-1, SMC-3, and one of 3 meiosis-specific kleisin subunits: REC-8 and the functionally redundant COH-3 and COH-4 [22–24]. Both REC-8 and COH-3/4 cohesin promote pairing and recombination between homologous chromosomes during early meiosis, thus ensuring chiasma formation. However, SCC appears to be provided by REC-8 complexes, while COH-3/4 complexes associate with individual chromatids [25,26]. Previous work indicated that *rec-8* single mutants have 12 univalents at meiosis I, with each pair of sister chromatids held together by recombination events dependent on COH-3/COH-4 cohesin [25,27]. Sister chromatids segregated equationally at anaphase I of *rec-8* mutants with half the chromatids going into a single polar body [23]. This suggests that *rec-8* embryos enter metaphase II with 12 separated sister chromatids. Although it was reported that *rec-8* embryos do not extrude a second polar body, the structure of the metaphase II spindle was not described in detail. To address the question of whether chromosomal bipolarity is required for spindle bipolarity, we first monitored metaphase II spindle assembly in a *rec-8* mutant by time-lapse imaging of living embryos *in utero*.

## Results

### Apolar spindles assemble around separated sister chromatids of metaphase II *rec-8* embryos

Time-lapse *in utero* imaging of control embryos with microtubules labelled with mNeon-Green::tubulin and chromosomes labelled with mCherry::histone H2b revealed bipolar spindles that shorten, then rotate, then segregate chromosomes in both meiosis I and meiosis II (Fig 1A and S1 Video). Wild-type embryos enter metaphase I with 6 bivalents and enter metaphase II with 6 univalents whereas *rec-8* embryos enter metaphase I with 12 univalents (Fig 1B and S1A Fig) and enter metaphase II with approximately 12 separated sister chromatids (Fig 1B) [23]. Time-lapse imaging of *rec-8* embryos revealed bipolar metaphase I spindles that shortened, rotated, and segregated chromosomes (Fig 1C, -1:45–5:15; and S2 Video). Metaphase II *rec-8* embryos, however, assembled an amorphous cloud of microtubules around separated sister chromatids which did not segregate into two masses. The apolar spindle shrank with timing similar to spindle shortening that occurs during wild-type meiosis (Fig 1C, 9:15–18:00). Because spindle shortening is caused by APC-dependent inactivation of CDK1 [28], this suggests that the failure in metaphase II spindle assembly is not due to a lack of cell cycle progression. The bipolar nature of metaphase I *rec-8* spindles and the apolar nature of *rec-8* metaphase II spindles was confirmed by time-lapse imaging of GFP::ASPM-1 (Fig 1D). ASPM-1 binds at microtubule minus ends [29] so the dispersed appearance of GFP::ASPM-1 on *rec-8* metaphase II spindles suggests that microtubules are randomly oriented in the spindle.

Time-lapse imaging of the kinetochore protein GFP::MEL-28 in *rec-8* embryos revealed metaphase I univalents with discrete bipolar structure similar to wild-type metaphase II

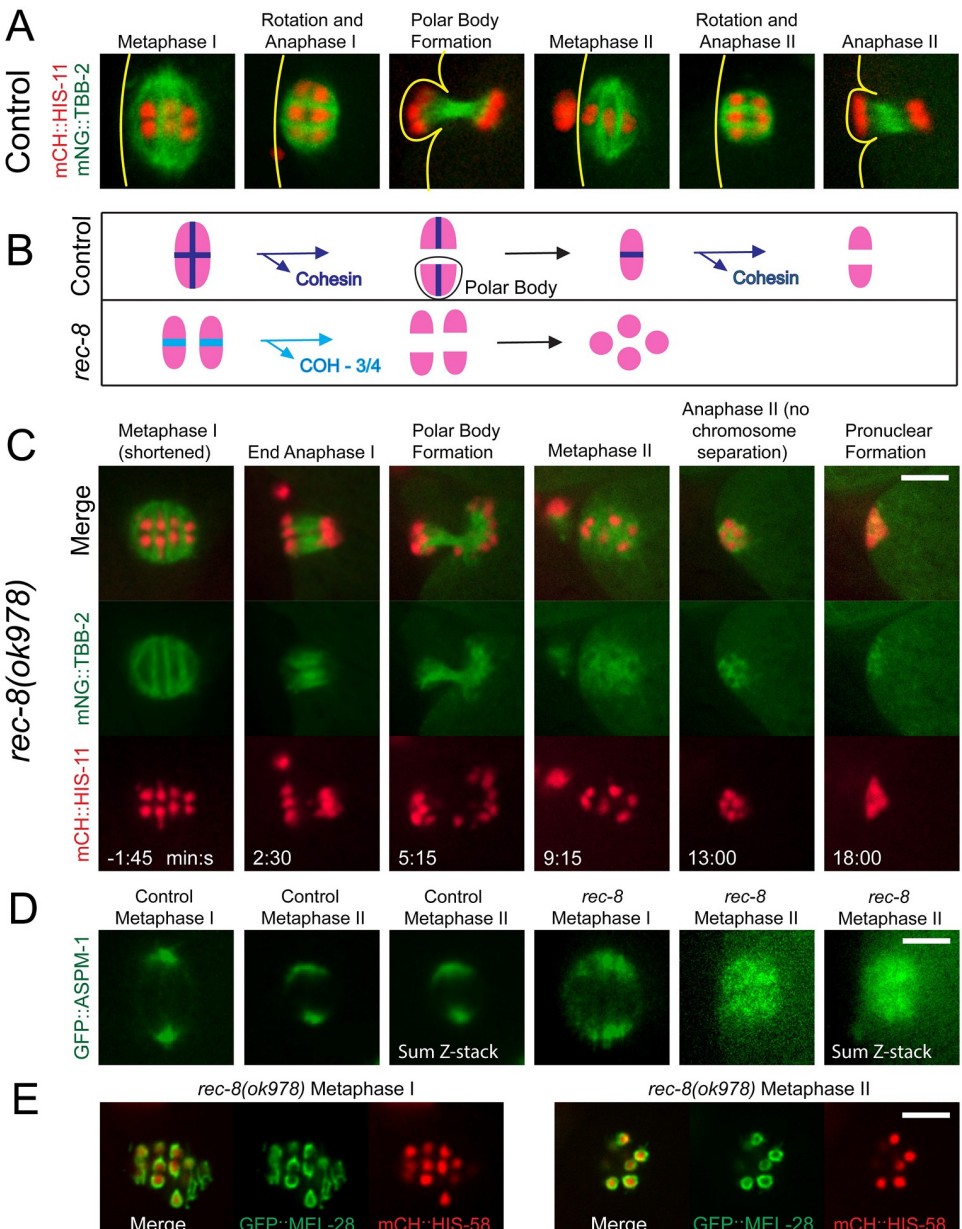

**Fig 1. Metaphase II spindles are apolar in *rec-8(ok978)*.** (A) In 8/8 control embryos, bipolar MI spindles shorten and rotate, chromosomes segregate, and a polar body forms. The cycle repeats with a bipolar MII spindle. Lines indicate the position of the cortex. (B) In metaphase I, both sister chromatids and homologs are bound by cohesin containing both REC-8 and COH-3/4 (dark blue); homologs are released and separate in Anaphase I; sister chromatids are released and separate in Anaphase II. In *rec-8(ok978)*, COH-3/4 cohesin (light blue) is present in an expanded region between sister chromatids in MI and no cohesin is present in MII. (C) Time-lapse imaging of *rec-8(ok978)* expressing mNG::TBB-2 and mCH::HIS-11. The metaphase II spindle appears disorganized and no anaphase chromosome separation occurs in 8/8 embryos. 0 minutes is the end of MI spindle rotation. (D) Control and *rec-8(ok978)* embryos expressing GFP::ASPM-1. Single-focal plane imaging was ended at metaphase II and z-stacks were acquired. 7/7 control MI spindles, 7/7 *rec-8* MI spindles, and 7/7 control MII spindles were bipolar. 8/8 *rec-8* MII spindles were apolar. (E) Imaging of *rec-8* embryos expressing GFP::MEL-28 revealed kinetochore cups in 4/4 MI spindles and chromatids enclosed by GFP::MEL-28 in 7/7 MII spindles. All bars = 4μm.

univalents, whereas metaphase II separated sister chromatids were enveloped by a continuous shell of GFP::MEL-28 with no bipolar symmetry (Fig 1E).

## Apolar spindles assemble around separated sister chromatids of metaphase I *rec-8 coh-4 coh-3* embryos and *spo-11 rec-8 coh-4 coh-3* embryos

To test whether the apparent inability of separated sister chromatids to drive bipolar spindle assembly is specific for meiosis II, we compared control embryos (Fig 2A and 2B) with embryos of a *rec-8 coh-4 coh-3* triple mutant which lack meiotic cohesin and therefore enter metaphase I with 24 separated sister chromatids [23] (Fig 2A and S1C Fig). In the majority of these embryos, an amorphous cloud of microtubules assembled around the separated sister chromatids (Fig 2C) at the same time after ovulation that a bipolar spindle assembled in control embryos (Fig 2B). This amorphous cloud shrank in diameter (Fig 2C, -0.20) at a similar time as control spindles, which shortened prior to anaphase chromosome separation (Fig 2B, -1:10). The mutant spindles did not undergo anaphase like control spindles. In a minority of *rec-8 coh-4 coh-3* triple mutant embryos, a bipolar metaphase I spindle started to form (Fig 2D, -6:50 and -6:10) but then quickly collapsed into an amorphous cloud of microtubules (Fig 2D, -4:20). These spindles also shrank with timing similar to wild-type spindle shortening and did not undergo anaphase (Fig 2D, 5:00). Triple mutant embryos assembled a second amorphous mass of microtubules at the time of normal metaphase II spindle assembly (Fig 2D, 13:20) and this meiosis II spindle also shrank without segregating chromosomes (Fig 2D, 16:10). Similar results were obtained by fixed immunofluorescence (Fig 2E).

We also examined meiotic embryos within a *spo-11 rec-8 coh-4 coh-3* quadruple mutant (Fig 3A), which lack meiotic cohesin and the double strand breaks that initiate meiotic recombination (*spo-11* mutation) and also enter metaphase I with 24 separated sister chromatids [23] (Fig 3B). In all of these embryos, an amorphous mass of microtubules formed around the 24 chromatids (Fig 3A, -2:30; and S3 Video). This cloud of microtubules shrank with similar timing to wild-type spindle shortening and was not followed by any separation of chromosomes (Fig 3A, -2:30–2:30). A second large mass of microtubules formed at the time that a metaphase II spindle normally forms (Fig 3A, 12:15). This metaphase II mass also shrank with similar timing to normal spindle shortening (Fig 3A, 12:15–16) and chromatids did not separate into two masses and polar bodies did not form in 10/10 time-lapse sequences. Possible reasons for the stronger defect in the quadruple mutant than the triple mutant are discussed below. These results indicated that bipolar spindle assembly around separated sister chromatids that lack both cohesin and cohesion, is severely defective at both metaphase I and metaphase II.

## Bipolar spindles assemble around separated sister chromatids of metaphase I *spo-11 rec-8* embryos

To distinguish whether cohesin vs cohesion is required for bipolar spindle assembly, we analyzed *spo-11 rec-8* double mutants (Fig 3C) which enter metaphase I with 24 separated sister chromatids [24] (Fig 3D and S1B Fig) but have been reported to retain COH-3/4 cohesin on pachytene chromosomes [24,26]. Bipolar metaphase I spindles assembled in *spo-11 rec-8* double mutants and these spindles shortened, rotated, and then segregated the chromatids into two masses (Fig 3C, -6:50–5:20; and S4 Video). During meiosis II, an amorphous mass of microtubules assembled around the chromatids and this mass shrank but did not separate chromatids into two masses (Fig 3C, 16:10–18:40), similar to meiosis I in the triple and quadruple mutant, and meiosis II in the triple mutant, the quadruple mutant and the *rec-8* single mutant. The spindle pole protein, GFP::MEI-1, clearly labelled two poles of metaphase I and

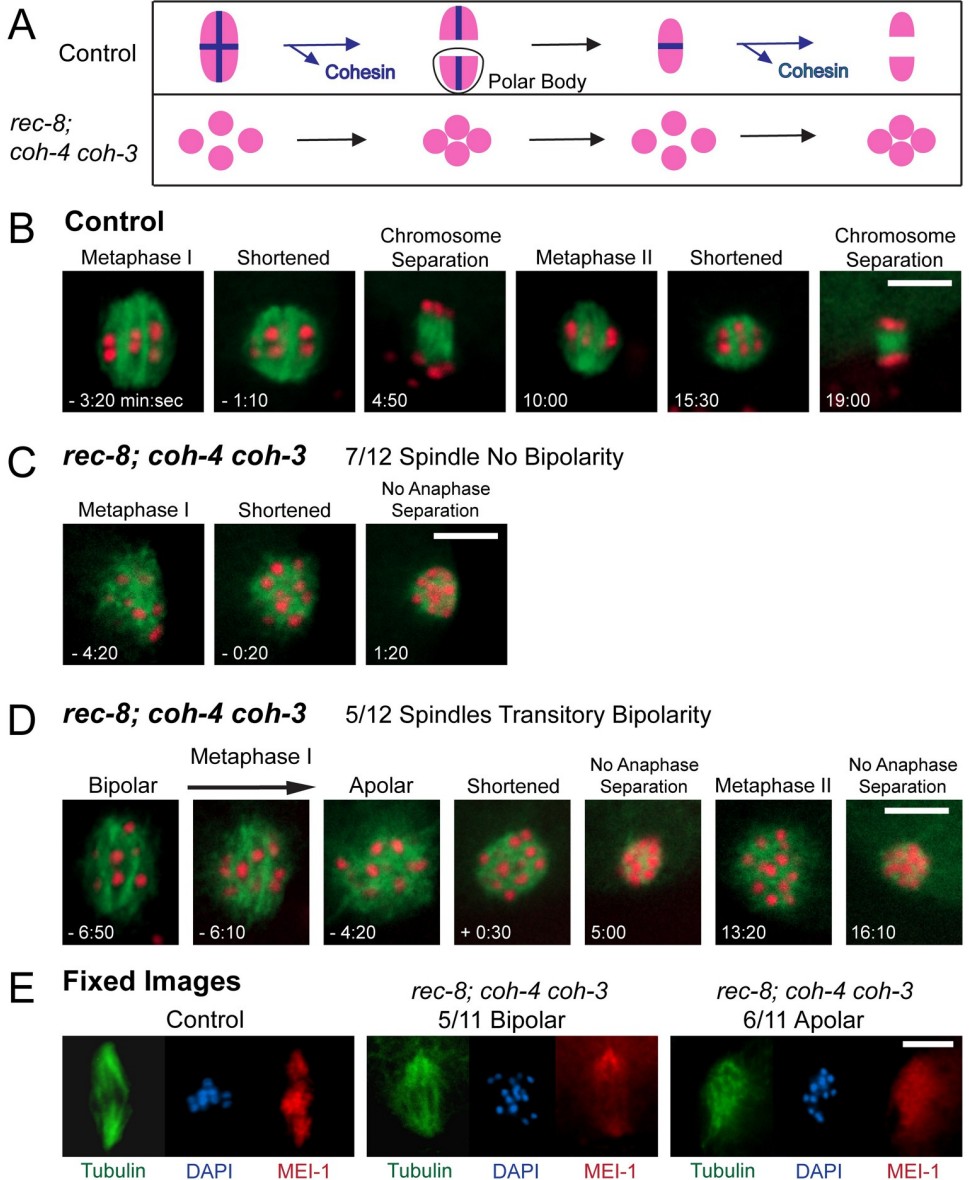

**Fig 2. Bipolar spindles can form in *rec-8; coh-4 coh-3* embryos, but they are unstable and become apolar.** (A) In metaphase I of control embryos, both sister chromatids and homologs are bound by cohesin; homologs are released and separate in Anaphase I; sister chromatids are released and separate in anaphase II. In *rec-8; coh-4 coh-3* embryos, no cohesin is present in either MI or MII and chromatids do not separate. (B) Time-lapse images of a control embryo expressing mNG::TBB-2 and mCH::HIS-11 show bipolar meiosis I and meiosis II spindles which shorten and undergo anaphase chromosome separation. Time 0:00 for B, C, and D is the time of full contact between the spindle and the cortex. (C) Time-lapse images captured with 7/12 *rec-8; coh-4 coh-3* embryos expressing mNG::TBB-2 and mCH::HIS-11 show MI spindles which were apolar at ovulation, then shortened and chromosome separation did not occur. MII (not shown) was similar to MI. (D) In 5/12 embryos, MI spindles initially appeared to be bipolar, but were unstable and became apolar. The MI spindles shortened and no anaphase chromosome separation occurred. MII was similar to MI. (E) Control and *rec-8; coh-4 coh-3* embryos were fixed and stained with both tubulin and MEI-1 antibodies and with DAPI. 10/10 Control and 5/11 mutant embryos had spindles with MEI-1 concentrated on chromosomes and at two poles. 6/11 mutant spindles were apolar. All bars = 5μm.

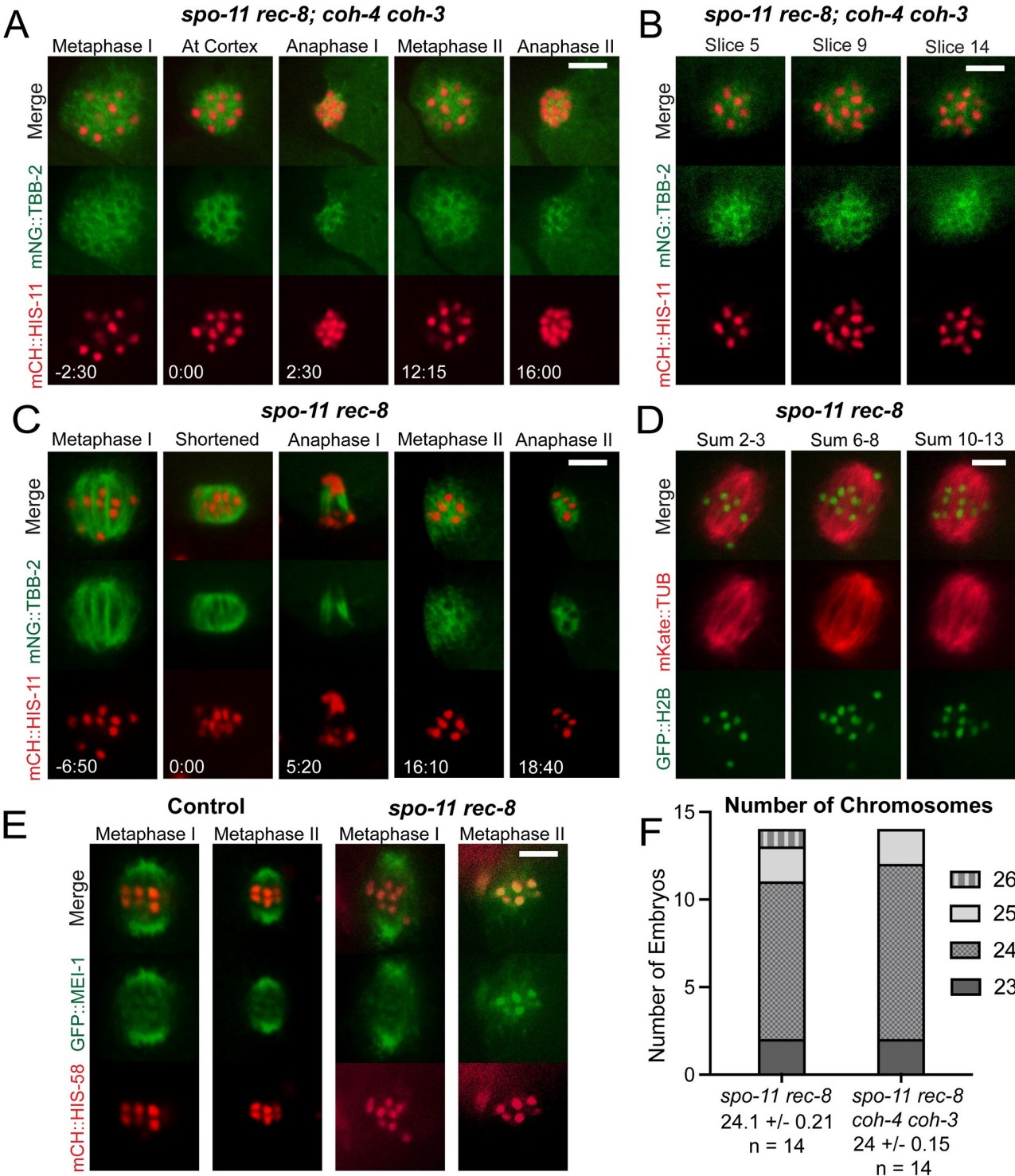

**Fig 3. *spo-11 rec-8; coh-4 coh-3* embryos have disorganized meiotic spindles whereas *spo-11 rec-8* embryos have bipolar spindles in meiosis I.** (A) Single-focal plane time-lapse imaging of a *spo-11 rec-8; coh-4 coh-3* mutant expressing mNeonGreen::TBB-2 and mCherry::HIS. Disorganized spindles were observed in both MI and MII in 10/10 embryos. 0 minutes is the time when the MI spindle contacts the cortex. (B) Z-stack slices of a *spo-11 rec-8; coh-4 coh-3* MI spindle show 24 chromatids with one chromatid visible in both slices 9 and 14. (C) Single-focal plane time-lapse imaging of 13/13 *spo-11 rec-8* embryos show bipolar MI spindles which undergo anaphase chromosome separation and MII spindles which are disorganized and do not undergo anaphase chromosome separation. 0 minutes is the completion of MI spindle rotation. (D) Combined z-stack slices of a *spo-11 rec-8* MI spindle show 24 chromatids. (E) Time-lapse imaging of *spo-11 rec-8* embryos expressing GFP::MEI-1. 10/10 Control MI spindles, 5/5 Control MII spindles and 9/9 *spo-11 rec-8* MI spindles were bipolar. 8/8 *spo-11 rec-8* MII spindles were apolar. (F) Graph showing chromosome numbers during MI in both *spo-11 rec-8*, and *spo-11 rec-8; coh-4 coh-3* mutant embryos. All bars = 4μm.

metaphase II control spindles but only labelled spindle poles of metaphase I *spo-11 rec-8* mutants (Fig 3E). GFP::MEI-1 was dispersed on metaphase II spindles, confirming the apolar structure of these spindles. GFP::MEI-1 also associated with chromosomes and this chromosome association was much more apparent in metaphase II *spo-11 rec-8* spindles (Fig 3E). However, the background subtracted ratio of mean GFP::MEI-1 pixel intensity on chromosomes divided by mean cytoplasmic intensity was not significantly increased between metaphase I and metaphase II for either *spo-11 rec-8* (MI: 7.01 ± 0.89, N = 5 embryos, n = 15 chromosomes; MII: 5.62 ± 0.76, N = 5, n = 15; p = 0.23) or control spindles (MI: 5.62 ± 0.33, N = 6, n = 18; MII: 5.47 ± 0.35, N = 6, n = 18; p = 0.74). This result indicated that the enhanced contrast of chromosomal GFP::MEI-1 in *spo-11 rec-8* embryos was due to the decrease in microtubule-associated GFP::MEI-1.

The ability of *spo-11 rec-8* embryos to form bipolar metaphase I spindles might be due to one or two univalents held together by residual COH-3/COH-4 cohesin. However, 24 chromosome bodies could be counted in Z-stacks of the majority of metaphase I spindles (Fig 3F) and all metaphase I spindles were bipolar (13/13 mNeonGreen tubulin, 9/9 GFP::MEI-1). The ability of *spo-11 rec-8* embryos to undergo anaphase I but inability to undergo anaphase II is consistent with the single polar body previously described for this double mutant [23].

## Cohesin rather than cohesion is required for bipolar spindle assembly

The ability of *spo-11 rec-8* mutants to build bipolar metaphase I spindles but not metaphase II spindles might be because metaphase I chromatids retain cohesin, as high levels of COH-3/4 associate with pachytene chromosomes of *rec-8* mutants [24,26]. This non-cohesive COH-3/4 cohesin might be removed at anaphase I, leaving the metaphase II chromatids with no cohesin. This hypothesis was validated by time-lapse imaging of the cohesin subunit, SMC-1::AID:: GFP, which would be a component of both REC-8 cohesin and COH-3/4 cohesin. SMC-1:: AID::GFP was found on control metaphase I and metaphase II chromosomes and on most metaphase I chromosomes of *spo-11 rec-8* mutants but was absent from the metaphase II chromatids of *spo-11 rec-8* mutants (Fig 4A–4C). The absence of SMC-1 from a subset of metaphase I *spo-11 rec-8* chromatids may be due to WAPL-1-dependent and WAPL-1-independent pre-anaphase removal pathways [25]. To more directly test the requirement for cohesin, we monitored metaphase I spindle assembly in embryos depleted of SMC-1 with an auxin-induced degron [30]. For this experiment we monitored endogenously tagged GFP:: LIN-5 as a spindle pole marker instead of GFP::ASPM-1 because the *aspm-1* gene is linked to *smc-1*. The majority of SMC-1-depleted embryos formed apolar metaphase I spindles (Fig 4D). The small number of multipolar spindles likely resulted from an incomplete depletion of SMC-1 as a subset of oocyte nuclei exhibited residual SMC-1::AID::GFP fluorescence after auxin treatment (S2A and S2B Fig) and auxin treatment only caused a reduced brood size (S1 Table) whereas null mutants have been reported to be completely sterile [31]. These results support the idea that cohesin on chromosomes rather than cohesion between chromosomes is required for bipolar spindle assembly during both meiosis I and meiosis II.

## A specific subclass of chromosome-associated Aurora B kinase correlates with competence for bipolar spindle assembly

We then asked why cohesin might be required for bipolar spindle assembly. During mitosis in cultured human cells [32] and fission yeast [33], cohesin-associated PDS5 recruits haspin kinase to chromosomes [32] and the recruited haspin phosphorylates histone H3 threonine 3. Although PDS5 has important functions during meiotic prophase in several species [34–37], a role in recruiting haspin during meiosis has not been reported to our knowledge. The survivin

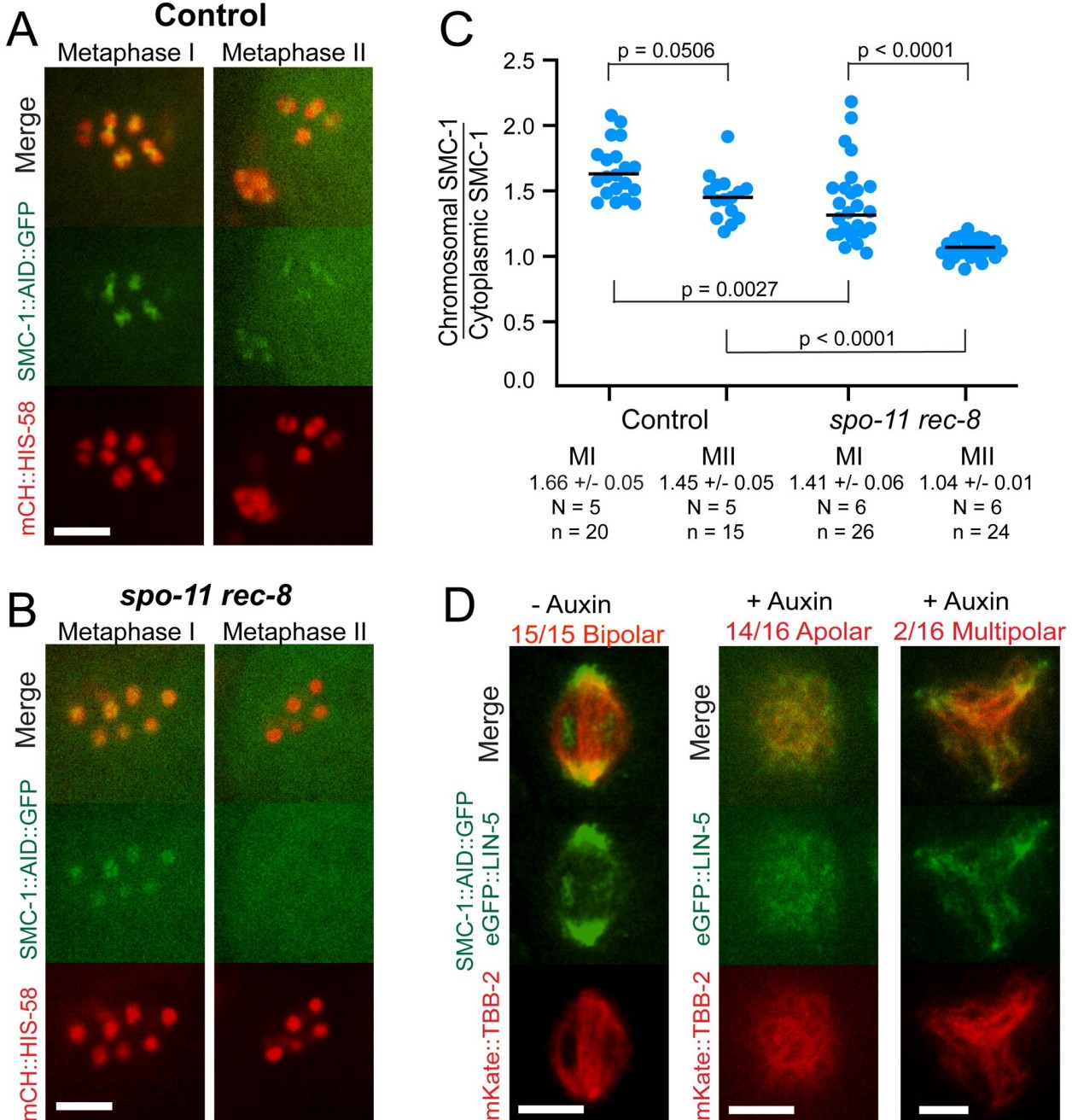

**Fig 4. Non-cohesive cohesin is sufficient for bipolar spindle formation.** Single-plane time-lapse images from control (A) and *spo-11 rec-8* (B) embryos expressing SMC-1::AID::GFP and mCH::HIS-58. (C) SMC-1::AID::GFP pixel intensities on individual chromosomes were determined relative to cytoplasmic background. N, number of embryos. n, number of chromosomes. (D) *C. elegans* expressing SMC-1::AID::GFP, eGFP::LIN-5 and mkate::TBB-2 were incubated overnight in the presence or absence of auxin. Single slices of z-stack MI images are shown. All bars = 4μm.

(BIR-1 in *C. elegans*) subunit of the CPC binds to the phosphorylated histone thereby recruiting Aurora B to chromosomes [32,38,39]. In *C. elegans*, haspin (HASP-1) is required to promote recruitment of Aurora B (AIR-2) to the midbivalent region in diakinesis oocytes [40] and AIR-2 is essential for bipolar meiotic spindle assembly in *C. elegans* [19,20]. Therefore we hypothesized that chromatids that lack cohesin-recruited AIR-2 would be unable to form

bipolar meiotic spindles. Time-lapse imaging of control embryos with endogenously tagged AIR-2::GFP (Fig 5A) revealed bright rings between homologs at metaphase I, microtubule association during anaphase I, bright rings between sister chromatids at metaphase II, and microtubule association during anaphase II as previously described [19]. In *rec-8* embryos, AIR-2 formed bright structures between sister chromatids at metaphase I and filled spaces between chromosomes at anaphase I, consistent with transfer to microtubules. However, at metaphase II in *rec-8* embryos, AIR-2::GFP was dim and diffuse on bipolar-spindle-incompetent separated sister chromatids, then became bright in regions between chromosomes, consistent with transfer to microtubules at anaphase II (Fig 5B). In *rec-8* embryos, AIR-2::GFP was significantly dimmer on chromosomes at metaphase II relative to metaphase I whereas no such decrease was observed in control embryos (Fig 5C).

In control -1 diakinesis oocytes, which will initiate meiosis I spindle assembly within 1–23 min [41], AIR-2::GFP brightly labeled the space between the homologous chromosomes in 6 bivalents. In contrast, GFP::AIR-2 was dim and diffuse on all of the bipolar-spindle-incompetent separated sister chromatids of *spo-11 rec-8 coh-4 coh-3* quadruple mutants (Fig 5D and 5E, S3 Fig). Unlike the quadruple mutant, a fraction of chromatids in the triple mutant had AIR-2::GFP intensities that overlapped with those of controls (Fig 5E) providing a possible explanation for the stronger spindle assembly defect in the quadruple mutant. Diakinesis oocytes of bipolar-spindle-competent *spo-11 rec-8* double mutants contained a mixture of separated sister chromatids with either dim diffuse AIR-2::GFP or bright patterned AIR-2::GFP (Fig 5D and 5E, S3 Fig). The bright patterned AIR-2::GFP on a subset of separated sister chromatids could also be observed in bipolar metaphase I spindles of *spo-11 rec-8* mutants (Fig 5F). The subset of metaphase I chromatids in *spo-11 rec-8* mutants with bright patterned AIR-2 was the same subset that retained COH-3/4 cohesin (S4 Fig). In bipolar-spindle-incompetent metaphase II embryos of *spo-11 rec-8* embryos, AIR-2::GFP was again dim and diffuse on all separated sister chromatids (Fig 5F). These results indicated that a specific subclass of AIR-2::GFP, that which is cohesin-dependent and forms a bright pattern on chromosomes, can promote bipolar spindle assembly. The subclasses of AIR-2::GFP that are cohesin-independent label chromatin dimly and diffusely, and label anaphase microtubules, but cannot efficiently promote bipolar spindle assembly.

To further test this idea, we analyzed sperm-derived chromatin in meiotic embryos. Whereas demembranated sperm [42] or DNA-coated beads [3] added to *Xenopus* egg extracts induce bipolar spindle assembly, the sperm-derived chromatin in *C. elegans* meiotic embryos does not induce spindle assembly [43]. Endogenously tagged GFP::SMC-1 was not detected on sperm-derived DNA in meiotic embryos (S5A Fig). When male worms with unlabelled AIR-2 were mated to hermaphrodites expressing endogenously tagged AIR-2::GFP, maternal AIR-2::GFP was recruited to the sperm DNA (S5B Fig) but this cohesin-independent AIR-2 did not induce bipolar spindle assembly. The cohesin-dependent subclass of AIR-2 might have a unique substrate specificity or it might be needed to reach a threshold of activity in combination with cohesin-independent AIR-2.

The reason for the heterogeneity of AIR-2 loading on separated sister chromatids of *spo-11 rec-8* mutants is not known, although it correlates with the heterogeneity of residual COH-3/4 cohesin (S4 Fig). The heterogeneity of AIR-2 loading on the 12 univalents of a *spo-11* single mutant correlates with heterogeneity in retention of LAB-1 and protein phosphatase 1, which remove haspin-dependent histone H3 T3 phosphorylation [40,44,45]. Our results suggest that bright patterned AIR-2 on only a subset of chromatids is sufficient to promote bipolar spindle assembly.

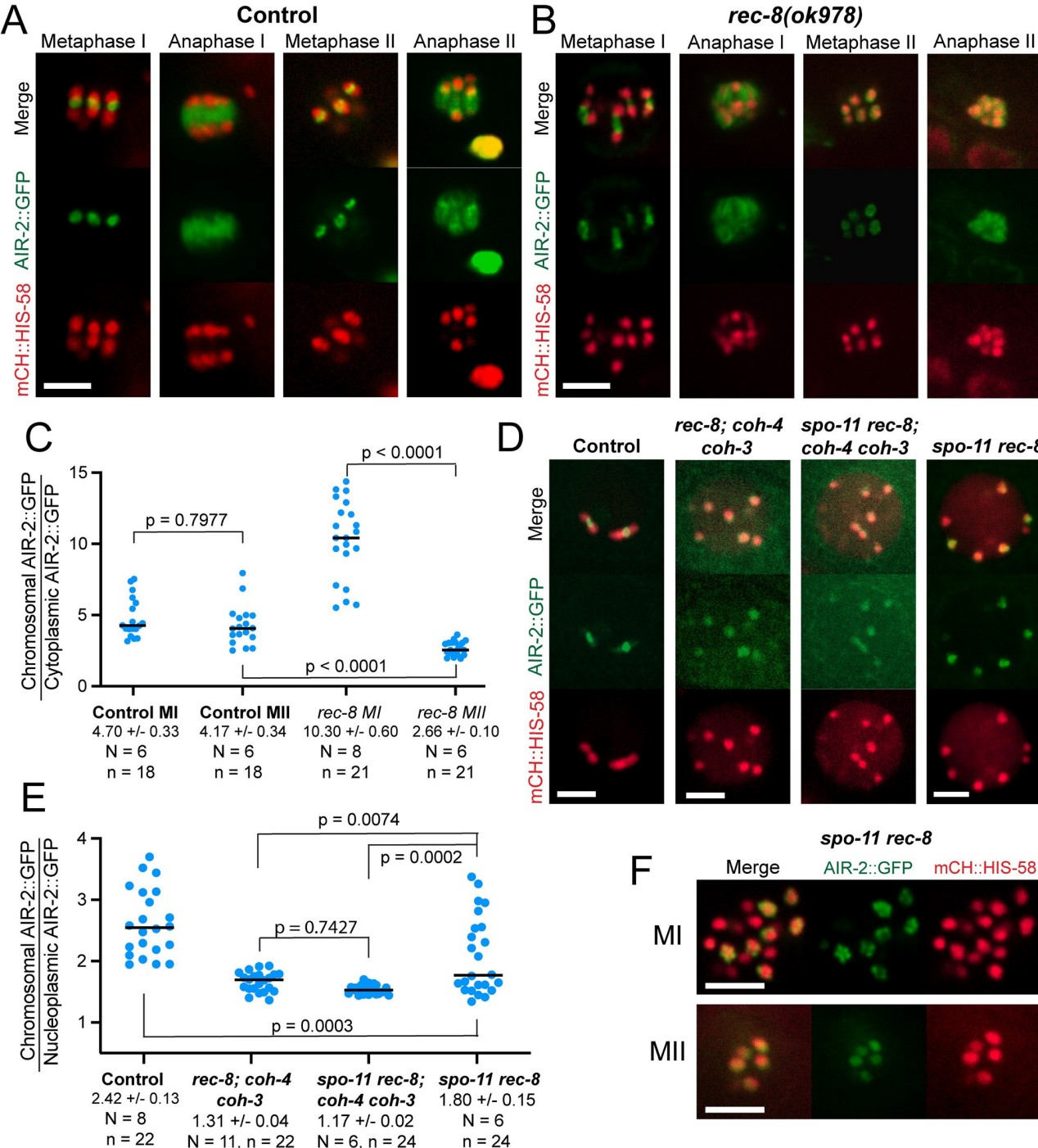

**Fig 5. AIR-2::GFP levels are diminished and diffuse in the absence of cohesin.** (A) In time-lapse images of control embryos, AIR-2::GFP is in the midbivalent ring structure during metaphase I and II and on MTs during anaphase I and II. (B) In *rec-8(ok978)*, AIR-2::GFP is an expanded ring structure during MI, diffuse on chromatids in MII and extends into a broader area during both anaphase I and II, consistent with transfer to microtubules. (C) Quantification of AIR-2::GFP intensities on chromosomes relative to the cytoplasm in control and *rec-8(ok978)*. Ratios varied depending on the distance of the chromosomes from the coverslip. N, number of embryos. n, number of chromosomes. The higher than control intensities in *rec-8* MI might be due to the previously reported [23] expanded ring structure between chromatids which might involve unresolved synaptonemal complex intermediates [25,27]. (D) -1 oocyte nuclei in living control and mutant worms expressing AIR-2::GFP and mCH::HIS-58. (E) Quantification of AIR-2::GFP intensities on chromosomes relative to the nucleoplasm in control and mutant oocytes. N, number of oocytes. n, number of chromosomes. (F) MI and MII metaphase chromosomes in living *spo-11(me44) rec-8(ok978)* embryos. All bars = 4μm.

## Haspin-dependent Aurora B kinase is required for bipolar meiotic spindle assembly

To more specifically identify the subclass of Aurora B that is required for bipolar spindle assembly, we analyzed a *bir-1(E69A, D70A)* mutant. This double mutation is equivalent to the D70A, D71A mutation in human survivin that prevents binding to T3-phosphorylated histone H3 and prevents recruitment of Aurora B to mitotic centromeres in HeLa cells [39]. Time-lapse imaging of mNeonGreen::tubulin in *bir-1(E69A, D70A)* mutants revealed apolar metaphase spindles that shrank without chromosome separation during both meiosis I and meiosis II (Fig 6A). The *bir-1(E69A, D70A)* embryos were unlike the cohesin mutants in that they entered meiosis I with 6 bivalents (11/11 z-stacks of -1 oocytes), suggesting successful formation of chiasmata between homologous chromosomes during meiotic prophase and intact SCC (Fig 6A). Endogenously-tagged AIR-2::GFP diffusely labeled both lobes of metaphase I (Fig 6B) and diakinesis (Fig 6C) bivalents in *bir-1(E69A, D70A)*. This was in contrast to the bright ring of AIR-2::GFP that is observed between the lobes in controls. AIR-2::GFP localized in a broader pattern consistent with transfer to microtubules during anaphase I and anaphase II (Fig 6B) as was observed in cohesin mutants. Apolar metaphase I spindles (Fig 6D, center) also formed after depletion of haspin kinase with an auxin-induced degron. Like *bir-1(E69A, D70A)* embryos, haspin-depleted embryos entered meiosis I with 6 bivalents (10/10 z-stacks of metaphase I), indicating the presence of chiasmata and SCC. As with cohesin mutants that were bipolar-spindle-incompetent, the fluorescence intensity of AIR-2::GFP on chromosomes was strongly reduced in both *bir-1(E69A, D70A)* and *hasp-1(degron)* embryos (Fig 6E). Whereas all *bir-1(E69A, D70A)* spindles were apolar, a minority of *hasp-1(degron)* spindles were multipolar (Fig 6D right, and 6F). Apolar spindles had undetectable phosphor H3 T3 staining whereas multipolar spindles had reduced phosphor H3 T3 staining relative to no auxin controls (S2C–S2E Fig). In addition, a low frequency of hatching was observed among the progeny of *hasp-1(AID)* worms on auxin plates (S1 Table). Because a *hasp-1(null)* mutant is completely sterile [46], the low hatch rate suggested that the low frequency of bipolar spindles in HASP-1-depleted worms was due to incomplete depletion by the degron. Because haspin is recruited to chromosomes by cohesin-associated PDS5 [32], these results indicated that the subclass of Aurora B that is recruited to chromosomes by cohesin and haspin-dependent phosphorylation of histone H3 is required for bipolar spindle assembly and that cohesin-independent and haspin-independent Aurora B on chromosome lobes and anaphase microtubules are not sufficient to drive bipolar spindle assembly.

## Cohesin is required for coalescence of microtubule bundles into spindle poles

*C. elegans* meiotic spindle assembly begins at germinal vesicle breakdown in the -1 oocyte that is still in the gonad. Microtubule bundles assemble within the volume of the nucleus as the nuclear envelope breaks down. Oocytes are then fertilized as they ovulate into the spermatheca and meiosis is completed in fertilized embryos that have moved out of the spermatheca into the uterus. Microtubule bundles can coalesce into poles either before, during, or shortly after ovulation [47–50]. The meiosis I spindle assembly defect in *spo-11 rec-8 coh-4 coh-3* mutants shown in Fig 3A was determined from time-lapse imaging of fertilized embryos in the uterus. To more precisely define the spindle assembly step that is defective in cohesin mutants, we conducted time-lapse imaging starting at nuclear envelope breakdown in -1 oocytes. In bipolar-spindle-competent control (Fig 7A and 7B) and *spo-11 rec-8* (Fig 7C and 7D) -1 oocytes, as well as bipolar-spindle-incompetent *spo-11 rec-8 coh-4 coh-3* (Fig 7E and 7F) -1 oocytes, microtubule bundles initially filled the entire volume of the germinal vesicle as it broke down.

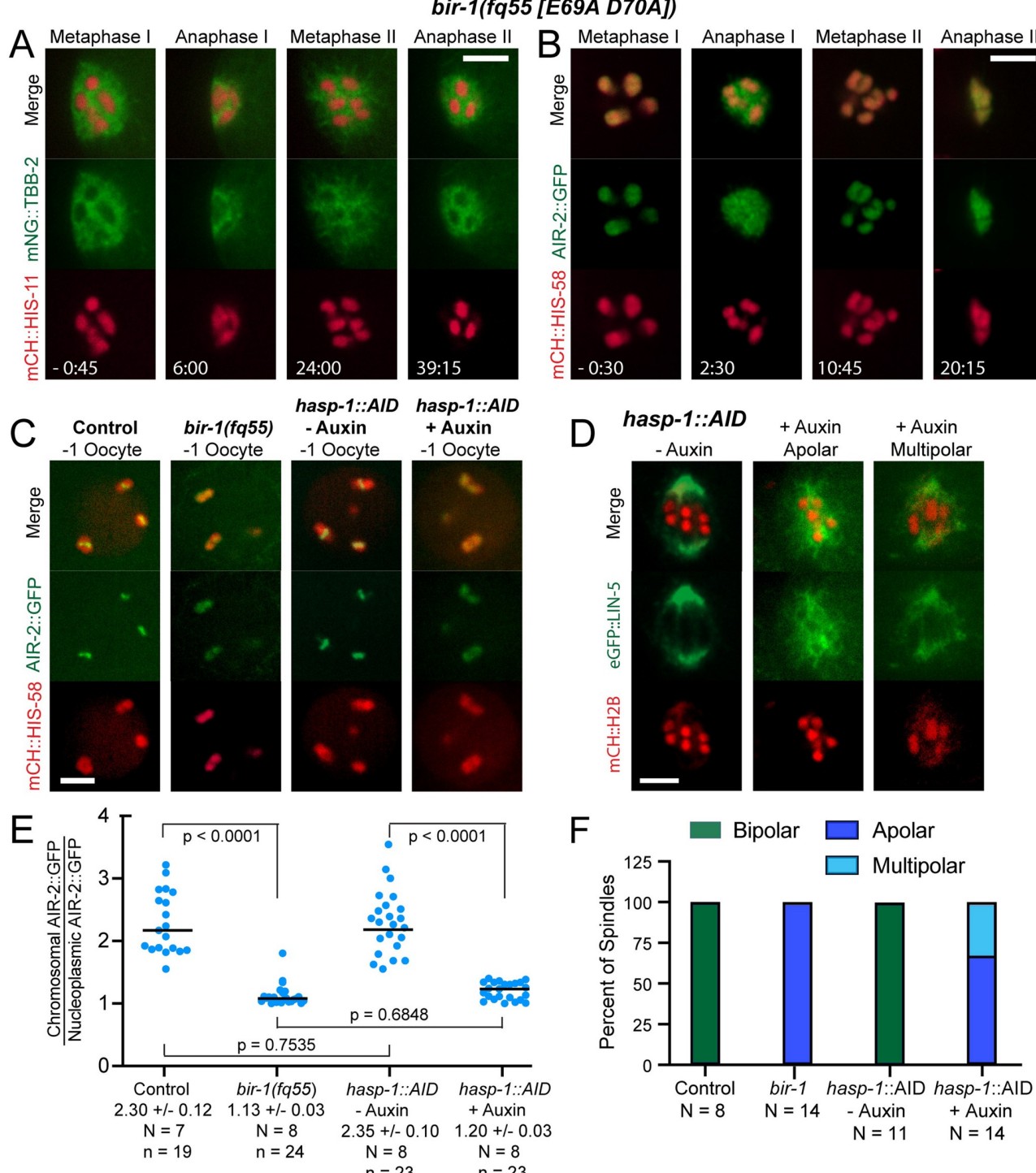

**Fig 6. AIR-2 is recruited by Survivin and Haspin for bipolar spindle formation.** (A) Time-lapse images of 14/14 *bir-1(fq55)* embryos expressing mNG::TBB-2 and mCH::HIS-11 show disorganized MI spindles and no MI anaphase chromosome separation. An example of a *bir-1(+)/bir-1(+)* control with the same transgenes is shown in Fig 1A. (B) Similar results were obtained in 4/4 *bir-1(fq55)* embryos expressing AIR-2::GFP, which is diffuse on both MI and MII metaphase chromosomes and present in a broader pattern consistent with microtubules during anaphase. An example of a *bir-1(+)/bir-1(+)* control with the same transgenes is shown in Fig 5A. *C. elegans*. (C) Single slices from z-stack images of -1 oocytes in *C. elegans* expressing AIR-2::GFP and mCH::HIS-58. 11/11–1 oocytes in *bir-1(fq55)* embryos had 6 mCH::HIS-58 labelled bodies. (D) Single-plane images of *hasp-1::AID* embryos expressing eGFP::LIN-5 and mCH::H2B. Left: Bipolar spindle without auxin (- auxin). Center: An apolar spindle with auxin (+ auxin). Right: A multipolar spindle with auxin (+ auxin). 10/10 MI spindles in Auxin-treated *hasp-1:::AID* embryos had 6 mCH::HIS-58 labelled bodies. (E) AIR-2::GFP pixel intensities on individual chromosomes were determined relative to nucleoplasmic background. N, number of oocytes. n, number of

chromosomes. (F) Graph showing percent of apolar, multipolar, and bipolar spindles in *bir-1* and auxin-treated *hasp-1::AID* embryos. N, number of embryos. All bars = 4 μm.

The microtubule bundles of control (Fig 7B) and *spo-11 rec-8* (Fig 7D) coalesced first into multiple poles, then into two poles as the oocytes squeezed into, then out of, the spermatheca. In contrast, the microtubule bundles of *spo-11 rec-8 coh-4 coh-3* (Fig 7F) did not coalesce even after ovulation into the uterus. This phenotype is consistent with that previously observed by fixed immunofluorescence of *air-2(degron)* embryos [20] and is distinct from the pole-stability phenotype reported for *zyg-9(RNAi)* where poles form but then fall apart [9]. In addition, the

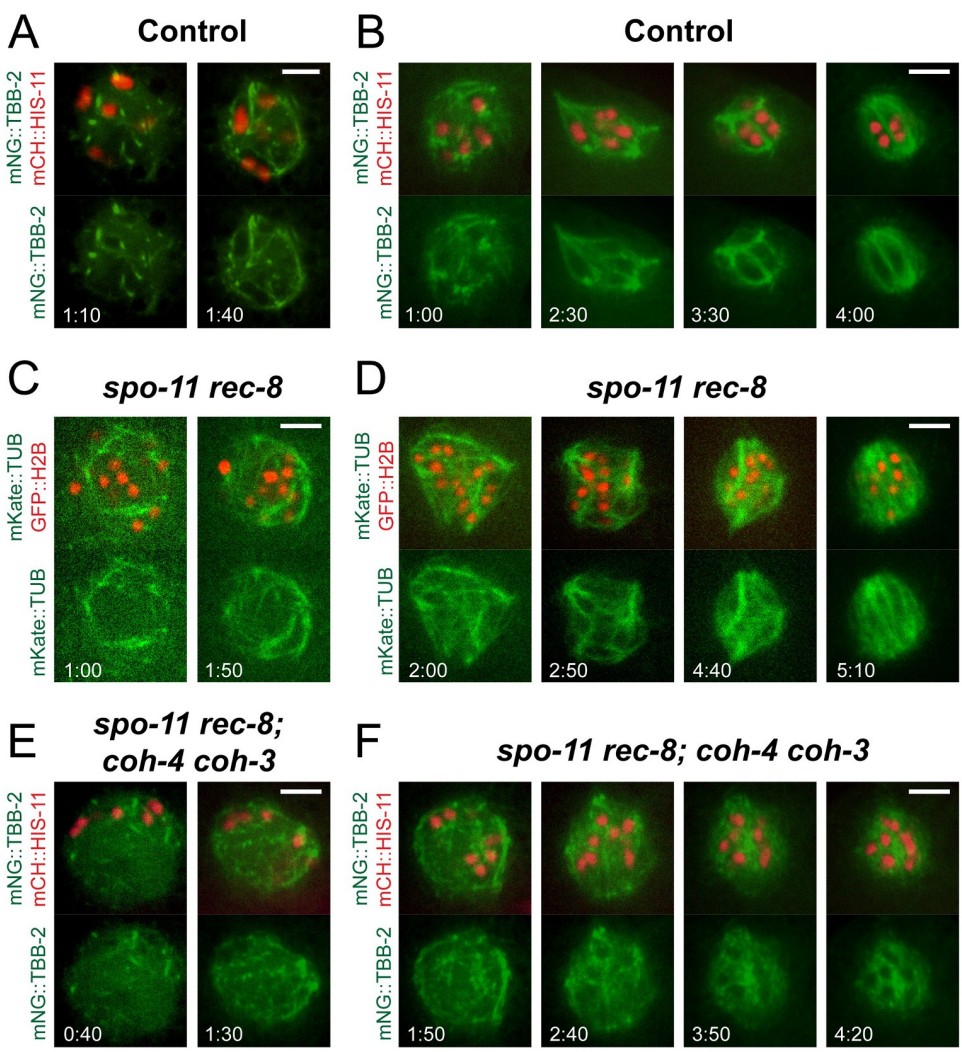

**Fig 7. Cohesin is necessary to direct the formation of spindle poles.** Time-lapse images in A, C, and E were captured in the gonad prior to ovulation; images in B, D, and F were captured post-ovulation in the uterus. (A, B) Time-lapse images of 7/7 control embryos show MT fibers organizing rapidly around chromosomes. Spindles become multipolar, then bipolar as poles coalesce. Times are from the initial observation of spindle MTs. (C, D) Time-lapse images of 13/13 *spo-11 rec-8* embryos show spindle fibers coalescing into multiple poles, then two poles. (E, F) Time-lapse images of 7/7 *spo-11 rec-8 coh-4 coh-3* embryos show that spindle fibers begin to form, but do not become organized into poles. All bars = 4μm.

mean fluorescence intensity of mNeonGreen::tubulin, indicative of microtubule density, was significantly reduced in apolar metaphase I spindles of *bir-1(E69A, D70A)* and *spo-11 rec-8 coh-4 coh-3* embryos relative to the bipolar spindles in control and *spo-11 rec-8* metaphase I spindles (S6 Fig). These results suggested that cohesin-dependent AIR-2 might regulate proteins that promote coalescence of microtubule bundles and promote microtubule polymerization, although other models are possible.

### Cohesin-dependent Aurora B kinase correlates with altered localization of spindle assembly factors on meiotic chromosomes

We hypothesized that cohesin-dependent Aurora B on chromosomes might activate microtubule-binding proteins that are required for coalescence of microtubule bundles and microtubule polymerization, or inhibit proteins that antagonize bundle coalescence and microtubule polymerization. Meiotic chromosome-associated spindle assembly factors include the katanin homolog, MEI-1 [51], the kinesin-13, KLP-7 [50,52], and the CLASP2 homolog, CLS-2 [19,53]. Loss of MEI-1 function results in apolar spindles with dispersed ASPM-1 [54] and reduced microtubule density [48,55] similar to those observed in cohesin mutants. However, apolar spindles in *mei-1* mutants are far from the cortex at metaphase I [47] whereas cohesin-mutant apolar spindles were cortical at metaphase I (Figs 1C, 1D, 2C, 3A and 3B). In addition, endogenously tagged GFP::MEI-1 was retained on chromosomes of apolar metaphase II *spo-11 rec-8* mutants (Fig 3E). These results suggest that MEI-1 is active in embryos that are deficient in cohesin-recruited AIR-2.

Endogenously tagged KLP-7::mNeonGreen localized to the midbivalent ring and to the two lobes of control bivalents (Fig 8A) but localized only to the two lobes in *bir-1(E69A, D70A)* mutants (Fig 8B). KLP-7 is also lost from the midbivalent ring after *air-2(degron)* depletion [20]. In living *spo-11 rec-8* double mutants, KLP-7::mNeonGreen localized in a bright pattern with a larger area on a subset of separated sister chromatids in bipolar-spindle-competent metaphase I embryos but labeled separated sister chromatids with a more uniform smaller area in bipolar-spindle-incompetent metaphase II embryos (Fig 8C and 8D). In fixed *spo-11 rec-8* embryos stained with antibodies and imaged with Airyscan, the pattern of KLP-7 on single chromatids was clearly distinct from that of AIR-2 (Fig 8E). In living *spo-11 rec-8* metaphase I embryos there was a positive correlation between the fluorescence intensity of endogenously tagged mScarlet::AIR-2 and the area of endogenously tagged KLP-7::mNeonGreen (Fig 8F). This result indicated that a subclass of bright patterned AIR-2 that is cohesin-dependent, and that correlates with bipolar spindle assembly, also correlates with a subclass of KLP-7 on chromosomes.

CLS-2::GFP labeled the kinetochore cups enveloping the two lobes of metaphase I bivalents but was not detected in the midbivalent region in control embryos (Fig 9A and S5 Video) in agreement with a previous study [56]. In contrast, CLS-2::GFP labeled kinetochore cups and the midbivalent region in *bir-1(E69A, D70A)* mutants (Fig 9A and S6 Video). In *spo-11 rec-8* double mutants, CLS-2::GFP localized in hollow spheres with a larger diameter on a subset of separated sister chromatids in bipolar-spindle-competent metaphase I embryos but labeled separated sister chromatids with more uniform, smaller diameter hollow spheres in bipolar-spindle-incompetent metaphase II embryos (Fig 9B and 9C). In *spo-11 rec-8* metaphase I embryos there was a weak positive correlation between the diameter of histone H2b and the diameter of CLS-2::GFP (correlation coefficient 0.37; Fig 9D) and a strong positive correlation (correlation coefficient 0.79) between the fluorescence intensity of endogenously tagged mScarlet::AIR-2 and the diameter of CLS-2::GFP spheres (Fig 9E and 9F). These results indicate that cohesin-dependent AIR-2 both excludes CLS-2 from the midbivalent region and

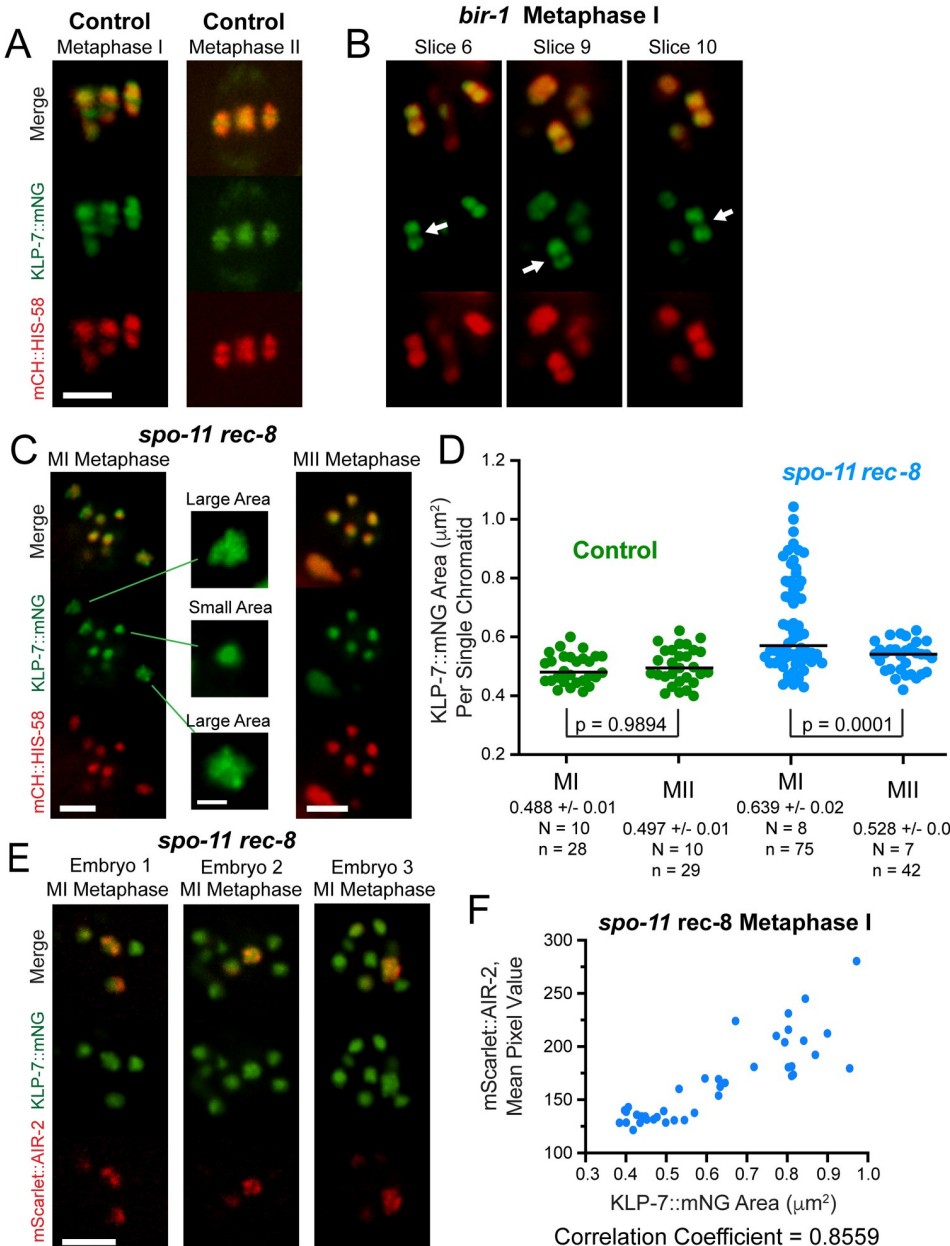

**Fig 8. Survivin-dependent AIR-2 is required for KLP-7 recruitment to the midbivalent ring.** (A) KLP-7::mNG localized to the two lobes and the ring complex (indicated by an arrowhead) of bivalents in 7/7 living control metaphase I embryos, but localized only to the two lobes in (B) 8/8 *bir-1(fq55) [E69A D70A]) embryos*. (Arrows indicate bivalents which clearly lack a ring complex). Bar = 3 μm. (C) In living *spo-11 rec-8* MI metaphase spindles, one subset of chromosomes had a small area of KLP-7::mNG and a second subset had a larger area of KLP-7::mNG that was unevenly dispersed around the DNA. Bar, spindle images = 3 μm. Bar, single chromosome images = 1 μm. (D) KLP-7::mNG areas were determined in *spo-11 rec-8* MI metaphase and MII metaphase spindles. N, number of embryos. n, number of chromosomes. (E) Single plane Airyscan images from z-stacks of 14/14 fixed *spo-11* rec-8 embryos showed expanded KLP-7::mNG on chromosomes with the highest mScarlet::AIR-2 fluorescence intensity. Bar = 3 μm. (F) Graph of mScarlet::AIR-2 mean pixel value relative to KLP-7::mNG area from live images. The Pearson r correlation coefficient is 0.8559. N = 8, n = 40, p < 0.0001.

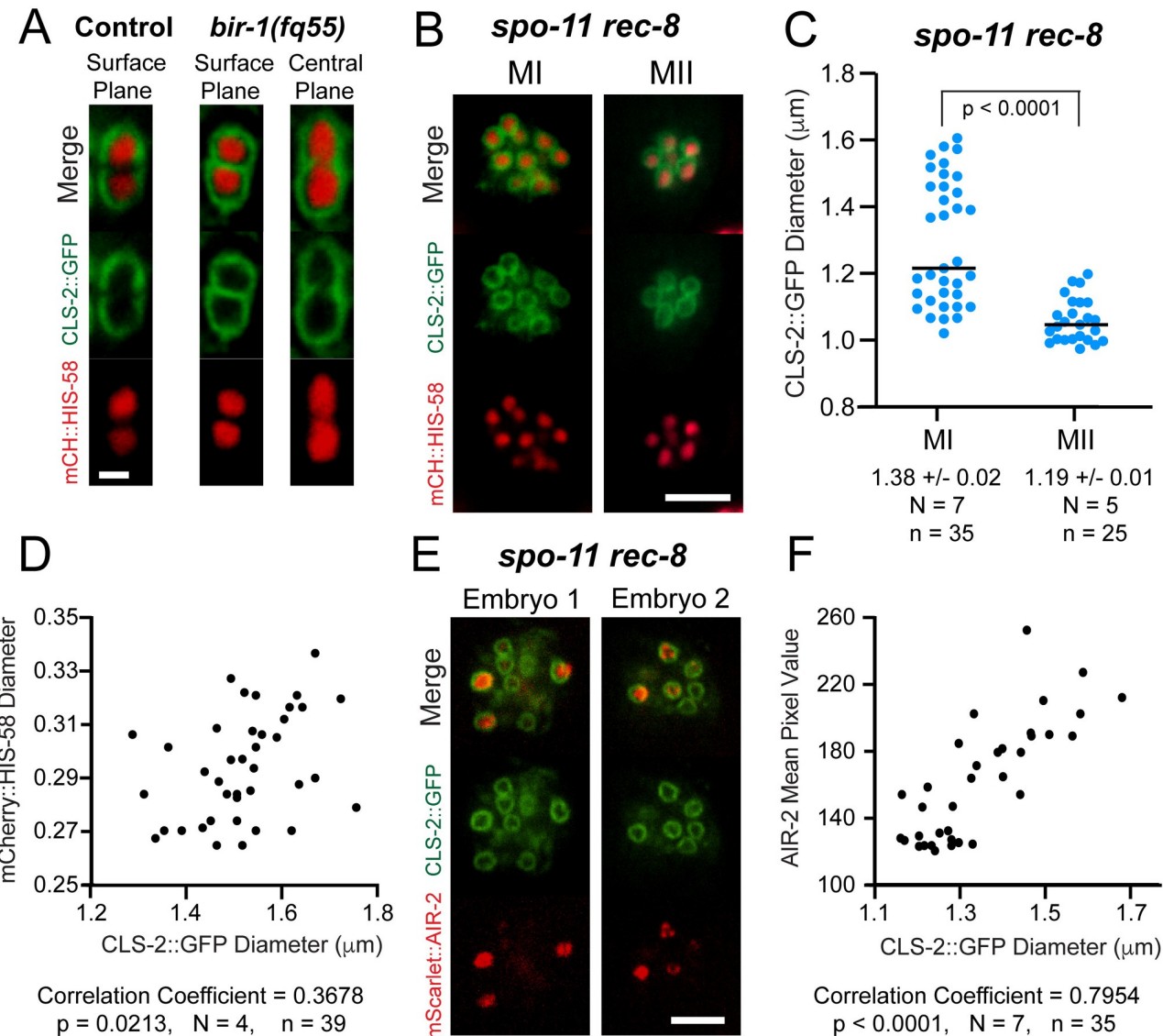

**Fig 9. BIR-1-recruited AIR-2 excludes CLS-2 from the midbivalent region.** (A) Individual chromosomes in living embryos expressing CLS-2::GFP show that CLS-2 is excluded from the midbivalent region in 9/9 control embryos and present in the midbivalent region in 16/16 *bir-1(fq55)* embryos. Bar = 1µm. (B) CLS-2::GFP encirles both MI and MII metaphase chromosomes in *spo-11 rec-8* embryos. (C) The diameter of CLS-2::GFP spheres on MI and MII metaphase chromosomes was determined. N, number of embryos. n, number of chromosomes. (D) Graph showing mCherry::HIS-58 diameter versus CLS-2::GFP diameter on *rec-8 spo-11* chromosomes. (E) Single plane images taken from z-stacks of two *spo-11 rec-8* embryos expressing CLS-2::GFP and mScarlet::AIR-2. (F) Graph showing mean pixel value of mScarlet::AIR-2 versus CLS-2::GFP diameter on *rec-8 spo-11* chromosomes. Bars, B and E = 4µm. N, number of embryos. n, number of chromosomes.

either recruits CLS-2 into larger spheres around separated sister chromatids or increases the diameter of separated sister chromatids.

## Most *C. elegans* meiotic SAFs are cytoplasmic

Vertebrate Ran-dependent SAFs bind importins through a nuclear localization signal (NLS) and are nuclear during interphase [5]. This arrangement places SAFs close to chromosomes at nuclear envelope breakdown when tubulin enters the nuclear space. In contrast, endogenously

GFP-tagged MEI-1, LIN-5, CLS-2, and AIR-1, which all contribute to bipolar spindle assembly in *C. elegans* [19,57–59], were all cytoplasmic before nuclear envelope breakdown (S7 Fig). In addition, KLP-15/16, which are required for spindle assembly, have been reported to be cytoplasmic in -1 oocytes [60]. The cytoplasmic location of these SAFs may sequester them from cohesin-associated CPC and thus prevent premature coalescence of microtubule bundles. These results also make it unlikely that most of the known SAFs during *C. elegans* meiosis are activated by the canonical Ran pathway, which involves release of an NLS from an importin by GTP-Ran [5].

## Discussion

Our results indicate that cohesin is required for efficient acentrosomal spindle assembly independent of its role in SCC because it is required for recruitment of a specific pool of Aurora B kinase to chromatin. The requirement for cohesin is independent of SCC because separated sister chromatids bearing COH-3/4 cohesin in *spo-11 rec-8* double mutants support the assembly of bipolar spindles. In contrast, separated sister chromatids in mutants lacking any cohesin assembled amorphous masses of microtubules with no discrete foci of spindle pole proteins. The cohesin-dependent pool of Aurora B kinase is then required for microtubule bundles to coalesce to form spindle poles during *C. elegans* oocyte meiotic spindle assembly. In the absence of either cohesin, haspin kinase, or phosphorylated histone H3-bound survivin, Aurora B remains dispersed on metaphase chromatin and localizes on anaphase microtubules but is insufficient to promote spindle pole formation. This could be due to a need for a threshold concentration of Aurora B on chromatin or a need for a specific activity unique to cohesin-dependent Aurora B.

The mechanism by which a specific pool of Aurora B kinase promotes spindle pole formation is not clear. In *Drosophila* oocytes, Aurora B phosphorylates the microtubule-binding tail of the kinesin-14, ncd, releasing it from inhibition by 14-3-3 [61]. Aurora B is thus activating ncd's ability to bind microtubules and its loss should have a similar phenotype to loss of ncd. In *C. elegans*, depletion of the kinesin-14's, KLP-15/16, results in apolar meiotic spindles [60], a phenotype similar to that reported for loss of Aurora B [20] or cohesin (this study). Thus KLP-15/16 are potential targets of activation by Aurora B in *C. elegans*. Completely apolar meiotic spindles have also been observed in *C. elegans* upon depletion of MEI-1/2 katanin [54] and AIR-1 [59]. Thus MEI-1/2 and AIR-1 are potential targets of activation in *C. elegans*.

In contrast with activation of kinesin-14, Aurora B promotes bipolar spindle assembly by phosphorylating and inhibiting the microtubule-disassembly activity of the *Xenopus* kinesin-13, XMCAK [62]. If Aurora B inhibits kinesin-13, then depletion of Aurora B or kinesin-13 should have opposite phenotypes. Indeed, depletion of kinesin-13 suppresses the spindle assembly defect of Aurora B inhibition in both *Xenopus* extracts [16] and *Drosophila* oocytes [63]. Loss of the *C. elegans* kinesin-13, KLP-7 [50,52] results in multi-polar meiotic spindles which might be viewed as the opposite phenotype of the apolar spindles resulting from depletion of AIR-2 [20] or cohesin (this study). Katanin is also inhibited in *Xenopus laevis* egg extracts by phosphorylation of an Aurora consensus site [64]. Whereas this exact site is not conserved in *C. elegans* MEI-1, the activity of MEI-1 is inhibited by phosphorylation at several sites [65]. If Aurora B acts by inhibiting one SAF, then over-expression of that SAF or expression of a non-phosphorylatable SAF should phenocopy loss of Aurora B. However, technical limitations of *C. elegans* transgene technology have limited over-expression of meiotic SAFs or expression of hyperactive mutant SAFs. If Aurora B acts by inhibiting or activating multiple SAFs, then reproducing the Aurora B depletion phenotype with phosphorylation site mutants of SAFs will be challenging.

Loss of haspin-dependent CPC in this study caused a change in the localization pattern of KLP-7 and CLS-2 on chromosomes. The CPC also regulates the chromosomal localization of the KLP-7 homolog, MCAK, on mammalian mitotic chromosomes [66]. Thus Aurora B may promote bipolar spindle assembly by regulating chromosomal targeting of SAFs in addition to regulating the activity of SAFs.

Depletion of SMC3, which should remove all cohesin from chromatin, has been reported in mouse oocytes [67] and *Drosophila* oocytes [68]. Metaphase I spindle defects were not reported in either case. In both cases, cohesin depletion may have been incomplete. Mouse oocyte spindle assembly is dependent on haspin [69], independent of Aurora B because Aurora A can substitute for B in the CPC [70], and dependent on Aurora A [71]. In *Drosophila*, bipolar spindle assembly is CPC-dependent [13] but the relevant CPC recruitment depends on borealin binding to HP1 [72] rather than survivin binding to haspin-phosphorylated histone H3. *Drosophila sunn* null mutants lack SCC and also form bipolar metaphase I spindles [68]. Thus it remains unclear how widely the cohesin-dependence of acentrosomal spindle assembly applies in phyla other than Nematoda. In addition, future analysis of centrosome-based *C. elegans* male meiosis in cohesin mutants should reveal whether the cohesin-dependence of spindle bipolarity is specific to acentrosomal spindle assembly.

Our time-lapse imaging revealed separated sister chromatids separating into two masses during anaphase I in *spo-11 rec-8* embryos. This result is consistent with the previously published observation of a single polar body and equational segregation interpreted from polymorphism analysis [23,24]. Similarly, *Drosophila sunn* mutants are able to carry out anaphase I [68]. HeLa cells induced to enter mitosis with unreplicated genomes likely have G1 non-cohesive cohesin on their individual unreplicated chromatids. These cells assemble bipolar spindles but do not separate the unreplicated chromatids into two masses. Instead, all of the chromatids end up in one daughter cell at cytokinesis [73]. In *C. elegans* meiosis, anaphase B occurs by CLS-2-dependent microtubule pushing on the inner faces of separating chromosomes [74]. During normal meiosis, the pushing microtubules assemble between homologous chromosomes in a manner that depends on the CPC which is localized between homologous chromosomes, thus driving correct chromosome segregation [19,20]. In a *spo-11 rec-8* double mutant, bright patterned AIR-2 is only on a subset of chromatids but microtubules still appeared to push all of the chromatids apart. Presumably, microtubules are pushing between any two chromatids. This *faux* anaphase likely occurs by the same mechanism as anaphase B in embryos depleted of outer kinetochore proteins [19,75].

The bipolar-spindle-competent separated sister chromatids of *C. elegans spo-11 rec-8* mutants had a severe congression defect (Fig 3C and 3D). In contrast, unreplicated chromatids in HeLa cells congress normally to the metaphase plate [73]. It is likely that antagonism between dynein in kinetochore cups and KLP-19 in the midbivalent ring is important for chromosome congression in *C. elegans* oocytes [76], thus the striking bipolar structure of *C. elegans* metaphase I bivalents and metaphase II univalents is essential for congression while dispensable for bipolar spindle assembly or anaphase.

## Materials and methods

CRISPR-mediated genome editing to create the *bir-1(fq55[E69A D70A])* allele was performed by microinjecting preassembled Cas9sgRNA complexes, single-stranded DNA oligos as repair templates, and dpy-10 as a co-injection marker into the *C. elegans* germline as described in Paix et al [77]. The TCGTACCACGGATCGTCTTC sequence was used for the guide RNA and the single-stranded DNA oligo repair template had the following sequence: tgtgcattttgcaacaaggaacttgattttgaccccgctgctgacccgtggtacgagcacacgaaacgtgatgaaccgtg.

*C. elegans* strains were generated by standard genetic crosses, and genotypes were confirmed by PCR. Genotypes of all strains are listed in S2 Table.

### Live *in utero* imaging

L4 larvae were incubated at 20˚C overnight on MYOB plates seeded with OP50. Worms were anesthetized by picking adult hermaphrodites into a solution of 0.1% tricaine, 0.01% tetramisole in PBS in a watch glass for 30 min as described in Kirby et al. [78] and McCarter et al [41]. Worms were then transferred in a small volume to a thin agarose pad (2% in water) on a slide. Additional PBS was pipetted around the edges of the agarose pad, and a 22-×-30-mm cover glass was placed on top. The slide was inverted and placed on the stage of an inverted microscope. Meiotic embryos or -1 diakinesis oocytes were identified by bright-field microscopy before initiating time-lapse fluorescence. For all live imaging, the stage and immersion oil temperature was 22˚C–24˚C. For all time-lapse data, single–focal plane images were acquired with a Solamere spinning disk confocal microscope equipped with an Olympus IX-70 stand, Yokogawa CSU10, Hamamatsu ORCA FLASH 4.0 CMOS (complementary metal oxide semiconductor) detector, Olympus 100×/1.35 objective, 100-mW Coherent Obis lasers set at 30% power, and MicroManager software control. Pixel size was 65 nm. Exposures were 300 ms. Time interval between image pairs was 15 s with the exception of Fig 6 images, which were captured at 10 s intervals. Focus was adjusted manually during time-lapse imaging. Control and experimental time-lapse data sets always included sequences acquired on multiple different days. For chromosome counting in oocyte nuclei, z-stacks were captured at 0.4 um intervals. For chromosome counting in metaphase spindles, z-stacks were captured at 0.2 um intervals. Chromosomes were counted in z-stacks, not in z projections.

### Timing

Control spindles maintain a steady-state length of 8 μm for 7 min before initiating APC-dependent spindle shortening, followed by spindle rotation and movement to the cortex [79]. Because the majority of our videos began after MI metaphase onset, we measured time relative to the arrival of the spindle at the cortex in Figs 1, 2, 3 and 6; for control embryos, this corresponded to the completion of rotation. For Fig 7, time was measured relative to the initial appearance of MT fibers.

### Fixed immunofluorescence and Airyscan imaging

*C. elegans* meiotic embryos were extruded from hermaphrodites in 0.8× egg buffer by gently compressing worms between coverslip and slide, flash frozen in liquid N2, permeabilized by removing the coverslip, and then fixed in ice-cold methanol before staining with antibodies and DAPI. The primary antibodies used in this work were mouse monoclonal anti-tubulin (DM1α; Sigma-Aldrich; 1:200), GFP Booster Alexa 488 (gb2AF488; Chromotek; 1:200), rabbit anti-GFP (NB600-308SS; Novus Biologicals; 1:600), rabbit anti-KLP-7 ([20]; 1:300), rabbit anti-MEI-1 ([80]; 1:200), rabbit anti-H3 pT3 (07–424; Merck Millipore; 1:700) and rabbit anti-COH-3 ([24];1:500). The secondary antibodies used were Alexa Fluor 488 anti-mouse (A-11001; Thermo Fisher Scientific; 1:200), Alexa Fluor 594 anti-rabbit (A11037; Thermo Fisher Scientific; 1:200) and Alexa Fluor Plus 647 anti-rabbit (A32733; Thermo Fisher). z-stacks were captured at 1-μm steps for each meiotic embryo using the same microscope described above for live imaging. Super resolution images shown in Fig 8E were acquired on a ZEISS LSM 980 with Airyscan 2.

## Auxin

*C. elegans* strains endogenously tagged with auxin-inducible degrons and a TIR1 transgene were treated with auxin overnight on seeded plates. Auxin (indole acetic acid) was added to molten agar from a 400 mM stock solution in ethanol to a final concentration of 4 mM auxin before pouring plates, which were subsequently seeded with OP50 bacteria. Depletion of SMC-1::AID::GFP is shown in S2A and S2B Fig. Depletion of HASP-1 was indicated by reduced phosphor h3T3 staining (S2C–S2E Fig). Bipolar spindle assembly occurs in *knl-1 (AID) knl-3(AID) tir1* worms [75] and in *dhc-1(AID)* worms (S8 Fig) treated with auxin using the same protocol. Bipolar spindle assembly also occurred in *smc-1::AID::GFP* worms with no auxin (Fig 4D) and *hasp-1(AID)* worms with no auxin (Fig 6D). Embryonic lethality was dependent on auxin for both degrons and auxin did not induce embryonic lethality in a strain carrying only endogenously tagged *lin-5* (S1 Table). Thus the spindle assembly defects observed for *smc-1::AID::GFP* and *hasp-1(AID)* likely do not result from non-specific effects.

## Fluorescence intensity measurements

Fluorescence intensity measurements are from single focal plane images chosen from z-stacks. Single focal plane images were chosen that had similar nucleoplasmic or cytoplasmic pixel values and in which the majority of a chromosome was in focus. A chromosome was judged to be in focus in the focal plane with the highest pixel intensity, largest diameter, and sharpest edges. Choosing focal planes with similar cytoplasmic or nucleoplasmic pixel values was used to partially eliminate the problem of spherical aberration due to different distances from the coverslip. For counting the number of bright vs dim AIR-2::GFP-labeled chromosomes in entire nuclei in S3 Fig, chromatids were subjectively scored as bright vs dim by comparing chromosomes within the same focal plane to compensate for the loss of intensity due to distance from the coverslip. In Figs 4B, 4C, 4E and 6E, total pixel values of chromosomal SMC-1::AID::GFP or AIR-2::GFP were obtained using the Freehand Tool (ImageJ software) to outline individual chromosomes. For each chromosome, the ROI was dragged to the adjacent nucleoplasm or cytoplasm and the total pixel value obtained. A background value was determined by dragging the ROI to a region of the image outside the worm. The values were background-subtracted, then divided in order to generate a ratio for comparison. This method was also used to determine the intensity of GFP::MEI-1 on chromosomes reported in the text of the results corresponding to Fig 3E. MEI-1 looks brighter on the chromosomes in the *spo-11 rec-8* metaphase II image because the original 16 bit image (65,000 grey levels) has been scaled to display the brightest pixel as 256 in the 8 bit (256 grey levels) figure panel. The chromosomes are not actually brighter as explained in the Results. In Fig 8D and 8F, areas of KLP-7::mNG on individual chromosomes was measured using the Freehand Tool (ImageJ). The diameter of CLS-2::GFP spheres in Fig 9 was calculated from the area using the equation $D = 2\sqrt{\frac{A}{\pi}}$, where D is diameter and A is area. Area was obtained by hand drawing a circular ROI over each sphere. Focal planes in which each sphere had the largest diameter were used. Mean mScarlet::AIR-2 pixel values in Figs 8 and 9 were determined after outlining individual chromosomes with the Freehand Tool (ImageJ). In S1 Fig, single-plane images were captured at the midsection of -1 oocytes. For each image, regions of nucleoplasm and cytoplasm were outlined and the mean pixel values determined. In S6 Fig, single-plane images were captured at the midsection of metaphase I spindles. For each image, mean pixel values of the spindle and a region of cytoplasm were determined. For both figures, the mean values were background-subtracted and divided to generate ratios for comparison.

## Statistics

P values were calculated in GraphPad Prism using one-way ANOVA for comparing means of three or more groups. Pearson correlation coefficients were calculated using GraphPad Prism.

## Supporting information

**S1 Fig. DNA body counts in -1 oocytes of mutant *C. elegans*.** (A) Single and Z-stack sum slices of a living *rec-8* oocyte nucleus expressing mCherry::HIS-11. *rec-8* oocyte nuclei contained 12.33 +/- 0.37 DNA bodies (n = 9), which included univalents and an occasional chromatid. (B) Single and Z-stack sum slices of a living *spo-11 rec-8* oocyte nucleus expressing GFP::H2B show 22 of the 24 total chromatids. *spo-11 rec-8* oocyte nuclei contained 23.8 +/- 0.01 DNA bodies (n = 10). (C) Single and Z-stack sum slices of a living *rec-8; coh-4 coh-3* oocyte nucleus expressing mCH::HIS-11. *rec-8; coh-4 coh-3* nuclei contained 24.5 +/- 0.43 DNA bodies (n = 14). 4/14 oocytes contained one or two small DNA bodies which may indicate chromosomes fragmented by SPO-11 activity. All bars = 5 μm.
(TIF)

**S2 Fig. Auxin depletion of SMC-1::AID::GFP and HASP-1::AID is incomplete in some embryos.** (A) Single-plane images of SMC-1::AID:GFP in the gonad of living worms incubated overnight in either the presence or absence of auxin. (B) The ratio of SMC-1::AID::GFP mean pixel intensity to mCH:HIS-58 mean pixel intensity in gonad nuclei was determined in worms incubated as described in (A). Several of the ratios in auxin-treated worms approach the values obtained in untreated worms. N, number of worms. n, number of nuclei. (C) Embryos from worms expressing HASP-1::AID and incubated in either the presence or absence of auxin were fixed and stained with tubulin and phosphor H3(T3) antibodies, and with DAPI. (D) Ratios of chromosomal to cytoplasmic H3(T3) antibody staining were determed in worms incubated as described in (C). N, number of spindles. n, number of chromosomes. (E) The values for worms incubated in the presence of auxin were separated into those obtained from chromosomes in apolar spindles and those obtained from chromosomes in multipolar spindles. N, number of spindles. n, number of chromosomes. All bars equal 4μm.
(TIF)

**S3 Fig. Some chromatids are bound by bright patterned AIR-2::GFP in *spo-11 rec-8* oocytes. (A)** Single chromosomes from z-stack images of living control and mutant *C. elegans* oocytes expressing mCherry::HIS-58 and AIR-2::GFP. Two examples are shown of a *spo-11 rec-8* chromosome, one bound by bright patterned AIR-2::GFP and one with dim diffuse AIR-2::GFP. All bars = 1μm. **(B)** Graph showing the percent of chromosomes bound by bright AIR-2::GFP in living -1 oocytes of control and mutant *C. elegans*. Z-stacks of entire nuclei were analyzed. For *spo-11 rec-8*, bright vs dim AIR-2::GFP was scored by only comparing chromatids within the same focal plane. Bright AIR-2::GFP was observed on 100 percent of control chromosomes, 0 percent of *spo-11 rec-8; coh-4 coh-3* chromatids and 39.5 +/- 4.0 percent of *spo-11 rec-8* chromatids. N, number of oocytes. n, number of chromosomes.
(TIF)

**S4 Fig. Colocalization of AIR-2 and COH-3 in *spo-11 rec-8* metaphase I embryos.** (A) Meiotic embryos within control and *spo-11 rec-8* worms expressing AIR-2::GFP were fixed and stained with DAPI, COH-3/4 antibodies, and GFP antibodies. The control spindle displays consistent intensities of AIR-2 and COH-3/4 on each chromosome while the *spo-11 rec-8* spindle displays varying intensities. Bars = 3 μm. (B) High magnification view of single chromatids from (A). The control chromosome shows bright COH-3/4 and bright AIR-2. Two

chromosomes from the same *spo-11 rec-8* embryo are shown, one with bright COH-3/4 and AIR-2 and one with dim COH-3/4 and AIR-2. Bars = 1 μm. (C) Graph showing mean pixel value of COH-3/4 versus mean pixel value of AIR-2 on *rec-8 spo-11* chromosomes. Mean pixel values were taken by using a circle ROI with a 22 pixel diameter (covering the entire univalent's area). N, number of embryos. n, number of chromosomes.
(TIF)

**S5 Fig. Maternal AIR-2, but not SMC-1, is recruited to the sperm DNA.** (A) Time-lapse images of 15/15 embryos from worms expressing SMC-1::GFP and mCH::HIS-58 in both oocytes and spermatocytes show no SMC-1::GFP on sperm-derived paternal DNA within the zygote during meiosis. SMC-1::GFP was observed in the sperm-derived paternal pronucleus in 7/7 embryos. Bar = 3 μm. (B) Male worms were soaked in mitotracker before mating to hermaphrodites. The sperm-derived paternal DNA is found at the center of the cloud of paternal mitochondria within meiotic embryos (far right). In 5/5 mated hermaphrodites, paternal AIR-2::GFP was present on spermatids, but was not detected post-fertilization within the cloud of paternal mitochondria in meiotic embryos identified by their position in the uterus adjacent to the spermatheca (+1 embryo). 13/13 unmated hermaphrodites expressing AIR-2::GFP, and 11/11 AIR-2::GFP expressing hermaphrodites mated with non-expressing males had AIR-2::GFP on the sperm DNA in +1 embryos. Bar = 4μm.
(TIF)

**S6 Fig. MT density is decreased in *spo-11 rec-8; coh-3 coh-4* and *bir-1(fq55)* spindles.** (A) Single slices from z-stack images of embryos expressing mNG::TBB-2 and mCH::HIS-11. Bar = 4μm. (B) Ratios of mean, background-subtracted mNG::TBB-2 pixel values in spindles vs. nearby cytoplasm of control and mutant embryos. N = number of embryos.
(TIF)

**S7 Fig. Spindle assembly factors are cytoplasmic prior to nuclear envelope breakdown.** (A) Single plane images of -1 oocytes in *C. elegans* expressing GFP::H2B, SMC-1::AID::GFP, and spindle assembly factors. Bar = 10 μm. (B) Nucleoplasmic to cytoplasmic ratios were determined for mean, background-subtracted pixel values in -1 oocytes.
(TIF)

**S8 Fig. Bipolar spindles form in the presence of Auxin.** *C. elegans* expressing DHC-1::AID::GFP, eGFP::LIN-5, mCH::H2B and mKate2::PH were incubated for 2–4 hours in the presence or absence of auxin. (A) Images of metaphase I spindles show that 9/9 spindles were bipolar in the absence of auxin and 10/10 metaphase I spindles were bipolar in the presence of auxin. (B) Quantification of spindle bipolarity. (C) Time-lapse images of *C. elegans* incubated in the absence of auxin show bipolar spindles shorten and rotate prior to chromosome separation (n = 5). (D) Time-lapse images of *C. elegans* incubated in the presence of auxin show bipolar spindles shorten and remain parallel to the cortex due to the depletion of DHC-1::AID::GFP (n = 7).
(TIF)

**S1 Table. Hatch rate data for auxin-induced degron experiments.**
(DOCX)

**S2 Table. *C. elegans* Strain List.** List of genotypes of all strains used in this paper.
(DOCX)

**S1 Data. Numerical values for all graphs shown in this paper.**
(XLSX)

**S1 Video. Metaphase I through anaphase II filmed in utero in a control strain.** Green is mNeonGreen::tubulin. Red is mCherry::histone H2b.
(MP4)

**S2 Video. Metaphase I through anaphase II filmed in utero in a *rec-8* strain.** Green is mNeonGreen::tubulin. Red is mCherry::histone H2b.
(MP4)

**S3 Video. Metaphase I through anaphase II filmed in utero in a *spo-11 rec-8 coh-4 coh-3* strain.** Green is mNeonGreen::tubulin. Red is mCherry::histone H2b.
(MP4)

**S4 Video. Metaphase I through anaphase II filmed in utero in a *spo-11 rec-8* strain.** Green is GFP::histone H2b. Red is mKate::tubulin.
(MP4)

**S5 Video. z-stack showing the pattern of CLS-2::GFP on control bivalents.**
(MP4)

**S6 Video. z-stack showing the pattern of CLS-2::GFP on *bir-1(fq55)* bivalents.**
(MP4)

## Acknowledgments

We thank Fede Pelisch, Arshad Desai, and the CGC, which is funded by NIH Office of Research Infrastructure Programs (P40 OD010440), for strains. We thank Sadie Wignall and Aaron Severson for antibodies. We thank Thomas Wilkop for assistance with Airyscan imaging.

## Author Contributions

**Conceptualization:** Karen P. McNally, Francis J. McNally.

**Data curation:** Karen P. McNally, Brennan M. Danlasky, Wenzhe Li, Francis J. McNally.

**Formal analysis:** Karen P. McNally, Elizabeth A. Beath, Brennan M. Danlasky, Ting Gong, Wenzhe Li, Francis J. McNally.

**Funding acquisition:** Enrique Martinez-Perez, Francis J. McNally.

**Investigation:** Karen P. McNally, Elizabeth A. Beath, Brennan M. Danlasky, Consuelo Barroso, Ting Gong, Wenzhe Li, Francis J. McNally.

**Methodology:** Karen P. McNally.

**Project administration:** Francis J. McNally.

**Resources:** Consuelo Barroso, Enrique Martinez-Perez.

**Supervision:** Enrique Martinez-Perez, Francis J. McNally.

**Validation:** Karen P. McNally.

**Visualization:** Karen P. McNally.

**Writing – original draft:** Karen P. McNally, Francis J. McNally.

**Writing – review & editing:** Karen P. McNally, Enrique Martinez-Perez, Francis J. McNally.

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
