## [Decision Letter · Decision Letter 0]

31 Mar 2022

Dear Dr McNally,

Thank you very much for submitting your Research Article entitled 'Cohesin is required for meiotic spindle assembly independent of its role in cohesion in C. elegans' to PLOS Genetics.

The manuscript was fully evaluated at the editorial level and by independent peer reviewers. The reviewers appreciated the attention to an important problem, but raised some substantial concerns about the current manuscript. Based on the reviews, we will not be able to accept this version of the manuscript, but we would be willing to review a much-revised version. We cannot, of course, promise publication at that time.

For resubmission I ask you to carefully address all reviewer’s comments. The reviewers are very clear in their comments as to which experiments are required, and which changes and additions to the text are requested.  This includes not only performing the required experiments and text edits, but also addressing low n values and missing information in the materials and methods section. I think it is possible to address these concerns in the timeframe suggested for revisions.

If you decide to revise the manuscript for further consideration at PLOS Genetics, please aim to resubmit within the next 60 days, unless it will take extra time to address the concerns of the reviewers, in which case we would appreciate an expected resubmission date by email to plosgenetics@plos.org.

[LINK]

We are sorry that we cannot be more positive about your manuscript at this stage. Please do not hesitate to contact us if you have any concerns or questions.

Yours sincerely,

Sarit Smolikove

Guest Editor

PLOS Genetics

Gregory P. Copenhaver

Editor-in-Chief

PLOS Genetics

Reviewer's Responses to Questions

**Comments to the Authors:**

Reviewer #1: Although cohesin is best known for its role in the sister chromatid cohesion (SCC) that holds replicated sister chromatids together from S phase until anaphase of mitosis and meiosis, the cohesin complex appears to perform other roles that are independent of its role in SCC. This manuscript from Frank McNally’s lab reports a novel SCC-independent role for cohesin in the establishment of bipolar, acentrosomal meiotic spindles. In many organisms, including C. elegans and mammals, oocytes lack centrosomes, and chromosomes appear to play a critical role in the formation of meiotic spindles. How spindle bipolarity is established in the absence of centrosomes remains poorly understood, but it has seemed likely that the back-to-back, bipolar arrangement of homologous chromosomes in meiosis I and of sister chromatids in meiosis II might be critical. By comparing spindle assembly in fertilized oocytes from C. elegans hermaphrodites that lack meiotic cohesin with those from mutants in which cohesin associates with meiotic chromosomes but does mediate SCC, the authors demonstrate that cohesin, but not SCC, is required for spindle bipolarity. As long as COH-3/4 cohesin is present on chromosomes, robust, bipolar spindles can form even when pairs of sister chromatids are detached from one another, indicating that back-to-back orientation of chromosomes is not needed for spindle bipolarity. Instead, COH-3/4 cohesin appears to recruit the Aurora B kinase AIR-2 to meiotic chromosomes through mechanism dependent on the Haspin kinase and survivin, and AIR-2 in turn appears to regulate the chromosomal association of several factors known to function in oocyte spindle assembly. Thus, this paper overturns a widely-held assumption regarding how spindle bipolarity is achieved, at least in C. elegans, provides new mechanistic insight into the process, and demonstrates a novel SCC-independent role for cohesin.

The experimental approach exclusively involves live imaging of microtubule and chromosome dynamics, as well as fluorescently-tagged versions of various microtubule and chromosome-associated proteins. When possible, quantitative data were extracted from the images. Although the methods used for the quantitation and certain technical aspects of the study need to be better described, the major conclusions are convincing and quite interesting. Some suggestions that may strengthen the conclusions or improve the readability of the paper follow.

Major Comments:

1. p.5, lines 19-20 and elsewhere: the term “single chromatid” is confusing. Phrases like “mutants that start meiotic spindle assembly with single chromatids” sound like there is only one chromatid. It might be clearer to say “detached sister chromatids” or “separated sister chromatids.”

2. Chromosome separation was not observed during anaphase I or II of spo-11 rec-8; coh-4 coh-3 quadruple mutants (p.8, lines 5-7 and 8-10), but it was observed during meiosis I of spo-11 rec-8 double mutants (p.8, lines 18-20). Since the authors likely have the data, it would be interesting to know whether polar body extrusion ever occurred in the quadruple mutant, and how often it occurred during meiosis I and II in the double mutant.

3. p.8, lines 21-23: “At metaphase II” is confusing. The mass of microtubules presumably assembled before metaphase II, while the failure to separate chromatids happened after metaphase II.

4. p.10, lines 1-4: “This non-cohesive COH-3/4 cohesin might be removed by separase at anaphase I, leaving the metaphase II chromatids with no cohesin. This hypothesis was validated by time-lapse imaging…” The imaging validated the hypothesis that cohesin was undetectable on metaphase II chromatids, but not that separase was required for removal of the COH-3/4 cohesin. This is likely true, but the authors should be careful not to suggest that it is proven by this experiment.

5. p.11, line 11 and throughout: The terms “spindle-incompetent” and “spindle-competent” are confusing if not misleading. Apolar spindles form around the detached sister chromatids of spo-11 rec-8; coh-4 coh-3 mutants (abstract line 10, first subheading in the results section, p.6 line 19, etc.). Thus, the mutants described as spindle-incompetent do not fail to form a meiotic spindle, they just do not form a bipolar spindle.

6. p.12, lines 2-4 and Fig. 4 D-F: in spo-11 rec-8 double mutants, some chromatids have very bright AIR-2 signal and others do not. Since COH-3/4 protein, but not SCC, appears to be required for the bright AIR-2, it would be quite interesting to know if the chromatids with intense AIR-2 signal also have high levels of COH-3/4.

7. pp.12-13: bir-1(E69A, D70A) mutants enter meiosis with 6 bivalents but form apolar spindles, and chromosomes do not separate in meiosis I or II. Is SCC released and do sisters disjoin in anaphase I or II in these mutants? If so, this should be shown. If not, this should also be shown, since it would increase the impact of this paper - SCC likely persists due to a failure to recruit AIR-2 to the region between homologs in meiosis I and between sisters in meiosis II, and therefore a failure to phosphorylate REC-8 to mark it for destruction by separase. Thus, the failure to segregate chromosomes could occur for very different reasons in the bir-1 mutant and in the spo-11 rec-8; coh-4 coh-3 mutants. From the figures, it does appear that disjunction fails, and this is quite intersting and critical to interpret the data.

8. p.14, lines 14-16: “These results suggested that cohesin-dependent AIR-2 regulates proteins that coalesce microtubule bundles and promote microtubule polymerization.” This model is consistent with the data, but it is not the only model consistent with the data. Thus, the statement as written is too strong.

9. p.14, lines 19-22: It is difficult to understand the hypothesis, since activating MT-binding proteins required for coalescence of MT bundles or MT polymerization and inhibiting such proteins should result in opposite phenotypes.

10. pp.16-17: the final section of the results does not fit with the rest of the paper and does not add much to the story. It is not obvious to me how the experiments “address the relative roles of the ran pathway and the CPC pathway,” as is claimed. Indeed, presence of SAFs in the cytoplasm does not preclude their regulation by Ran. The section should be rewritten to clarify the rationale and avoid overstating the conclusions, or perhaps the authors may wish to remove it since doing so would not significantly weaken the paper.

11. p.17, lines 19-21: It does not seem likely that the removal of residual cohesin at anaphase II would prevent the formation of a metaphase III spindle between meiosis II and the first mitotic cell cycle, because spindles DO form in the absence of cohesin during meiosis I and II of spo-11 rec-8; coh-4 coh-3 mutants – they just remain apolar.

12. As written, the manuscript suggests that the SCC-independent role of cohesin in bipolar meiotic spindle assembly is unique to the acentrosomal spindles that form in oocytes. This is likely true; however, it is formally possible that cohesin also plays an SCC-independent role that is required for formation of the centriolar meiotic spindles that form during spermatogenesis. It would be easy to look at spindle formation during spermatogenesis in spo-11 rec-8 and bir-1(E69A, D70A) mutants compared to spo-11 rec-8; coh-4 coh-3 mutants to see of loss of SCC or loss of cohesin disrupts spindle polarity. At the very least, this should be addressed in the discussion.

13. p.18: the long discussion paragraph on this page jumps from topic to topic and is somewhat difficult to follow. The possibility that Aurora B inhibits a target SAF comes out of the blue. After several examples of Aurora B-dependent inhibition of SAFs, the model is raised that the CPC might affect spindle pole assembly through activation or inhibition of multiple SAFs. Are there known examples of activation? The bir-1 mutants fails to accumulate KLP-7 in midbivalent rings and fails to exclude CLS-2 from a similar region, so there are clearly effects on protein localization – is this what the authors mean by activation and inhibition, or are they referring to enzymatic activities? The authors cite published data showing that depletion of either protein results in multipolar spindles and mention that effects of overexpression are not known. It is difficult to disentangle this data and understand what the model is. The paragraph needs to be revised, and a model figure would be very helpful.

14. p.18, lines 22-23 and p.19 lines 1-4: the fact that metaphase I defects were not observed following SMC3 depletion in Drosophila means that one cannot conclude that the bipolar spindle assembly observed in Drosophila sunn mutants suggests an SCC-independent requirement for cohesin in spindle assembly. It could simply be that cohesin is not required for bipolar spindle assembly in flies. Leading off with the sunn phenotype is somewhat misleading. This phenotype could still be mentioned, but it should come later in the paragraph after the SMC3 depletion result is described.

15. p.22, Auxin methods: more detail regarding the auxin experiments should be included. Was the auxin added to the molten agar before plates were poured, or was it added to plates after they were poured? What volume of auxin was added, or what concentration was the stock? What solvent was the auxin dissolved in? Were controls done to demonstrate that the phenotypes were a result of degradation of the target and not due to the solvent? Did the authors use IAA or NAA or one of the newer synthetic auxins? Degron experiments have given variable results in different labs, so it is critical to give detailed methods for these experiments.

16. pp.22-23, Fluorescence Intensity Measurements: insufficient detail is given for the quantification. In particular, what was used as the reference image for background subtraction (line 16)? Was it dark noise of the camera, or a region of the image outside of the worm, or something else? In lines 18-19, how was the diameter of CLS-2::GFP rings calculated from the area inside of an ellipse? I don’t understand how this would be done, since the angle of the ring relative to the objective would vary. More detail, and ideally the equation used, needs to be given. For intensity measurements done on single planes, what, if anything was done to ensure that the image plane was in the center of a chromosome or spindle and that differences in intensity were not simply due to the focal plane being analyzed? For supplemental figures 1 and 2, how were the regions of the spindle or nucleoplasm and cytoplasm selected? Were they the same size and shape?

17. Fig. 4C, legend: “Ratios varied depending on the distance of the chromosomes from the objective.” Presumably, this should say the distance of the chromosomes from the coverslip, since the distance between the objective and the focal plane should not change but chromosomes deeper within the gonad will be dimmer. If this is not the intended meaning, it should be stated more clearly. What is the evidence that the observed differences in this figure and others are not simply a consequence of distance to the chromosomes? Perhaps this needs to be discussed in the methods.

18. Fig. 7B, please use symbols to indicate bivalents in each subset of chromosomes described in the legend.

19. Supplemental Figure 1: In panel A, “Paternal DNA” is confusing – is this from a male, or simply the sperm DNA in an unmated hermaphrodite? In panel B, “+1 embryo” is never defined. What is the importance of the mitotracker label, and what is the conclusion from the two panels on the left? It is unclear in these panelswhere the sperm DNA is, if it is shown at all.

20. It has previously been shown that the axial element proteins HTP-3, HIM-3, and HTP-1/2 and the SC central region proteins form polycomplexes in rec-8; coh-4 coh-3 mutants but associate with chromosomes in spo-11 rec-8 mutants. Thus, AE formation requires cohesin but not SCC, similar to bipolar spindle assembly. It would be interesting to know if AEs are required for bipolar spindle assembly in spo-11 rec-8 mutants. If so, it would add another layer of mechanistic insight.

Minor comments, wording suggestions, etc.:

1. The authors should be consistent in their use of genetic nomenclature. In the standard nomenclature for C. elegans, gene names are separated by a space when on the same chromosome and by a semicolon when on different chromosomes. In the abstract (p.2, lines 7-8) commas separate both genes on the same chromosome and genes on different chromosomes (spo-11 and rec-8 on chromosome IV, coh-4 and coh-3 on chromosome V). Elsewhere (e.g. p.7, lines 21-22) no punctuation is used in the same genotype. While the authors may choose not to follow the standard nomenclature in the text to enhance the readability, they should strive to be consistent.

2. Standard nomenclature should be used in the Strain List. Mutations should be listed in order of their genomic location, from left to right on chromosome I, then chromosome II, etc. The allele designation for Karen Oegema’s lab is lt (LT), not it. Thus, their transgenes begin with ltIs or ltSi (e.g. ltIs37, the widely used mCherry-tagged histone).

3. p.5, line 22: “highly identical” is confusing. Things are identical or they are not.

4. p.6, lines 11-12: Severson et al (2009) did show in a supplemental figure that apolar spindles assemble around the chromatids of rec-8 embryos during meiosis II.

5. p.7, line 19: perhaps “continuous” instead of “contiguous.”

6. p.10, line 5: typo SMC-1::AAID::GFP

7. p.11, line 18: Is GFP::AIR-2 correct? AIR-2::GFP is used elsewhere.

8. p.12, lines 6-9: this should refer to Fig. S1, not Fig. S3.

9. p.13, line 23: the -1 oocyte has been cellularized and therefore is no longer part of the syncytial gonad.

10. Many citations throughout the text are formatted incorrectly. One example: “(K. McNally, Audhya, Oegema, & McNally, 2006; Srayko, O'toole, Hyman, & Müller-Reichert, 2006)”

11. Fig. 1A: it would be helpful to add timestamps similar to those shown in stills from other timelapses.

12. Fig. 2B: the legend states that there are 24 chromatids with one chromatid visible in both slices 9 and 14. This chromatid should be somehow indicated in the figure, for example with an arrowhead.

13. Fig. 3: it is confusing to have two figure panels labeled A.

14. Fig. 6A,B: it would be helpful to add symbols to indicate spindle poles in multipolar and bipolar spindles, as well as some examples of MT fibers that are referred to.

15. Fig. 6 legend: the statement “Images in E and F have been pseudocolored for increased clarity” is confusing. Aren’t all of the images in the paper pseudocolored, since they were captured with a greyscale camera?

16. Fig. 7A, it would help to have arrowheads pointing to the rings.

Reviewer #2: The manuscript by McNally et.al. assesses the contribution of chromosome structure to meiotic spindle assembly, by investigating whether the bipolar structure of C. elegans chromosomes is required for spindle assembly. They therefore analyze a set of mutants with defects in sister chromatid cohesion and demonstrate that in the absence of all cohesion (spo-11, rec-8, coh-3, coh-4), spindles failed to form. However, spo-11 rec-8 mutants formed bipolar spindles in MI, presumably due to the presence of residual COH-3/4-containing cohesin complexes. In these mutants, a subset of chromosomes are able to pattern AIR-2, and chromosomes with patterned AIR-2 have a different localization of spindle factors MCAK and CLS-2. Therefore, the authors conclude that cohesin is essential for spindle assembly independent of its role in sister chromatid cohesion.

This manuscript reports interesting findings that have the potential to be of interest to the field. However, there are some major issues that need to be corrected prior to publication.

Major points:

1. The authors argue that the spo-11 rec-8 mutant has 24 chromosomes, contrary to a prior report that argued that these mutants have 12 univalents that segregate equationally in MI (Figure 2A in Severson 2009). However, the data presented to support this conclusion (Figure 2D, 2F) is not convincing. The chromosome counting appears to have been done on metaphase-stage spindles, where the chromosomes are very close together (however, there are no details in the materials and methods describing this counting, so it is hard to tell; see point #2 below). If counting was done in metaphase, it would be difficult to definitively state that two chromosomes close together are not part of the same bi-lobed univalent (as is seen in rec-8 single mutants, and as Severson argued also occurs in rec-8 spo-11). This is especially true because it has been shown that the chromatids in rec-8 mutants are quite separated, even though they are still held together (the authors also show this in the current manuscript, Figure 4B, Metaphase I image). Therefore, the 24s counted in Figure 2F could actually be 12s, just with the individual sisters further away from each other than would be seen in WT MII univalents. If the authors want to argue that this mutant has 24 individual chromosomes, they need to demonstrate this more rigorously. They should count chromosomes in diakinesis, where the chromosomes are more spread out, and also image additional markers to aid in determining if particular chromosomes are linked to others. I suggest imaging MEL-28, as they do in Figure 1E, and also AIR-2, as they do in Figure 4B, which may reveal chromosome connections; this imaging should be done using multiple z-stacks, to make sure they are imaging all chromosomes (unlike the single-slice images shown in many other panels, such as Figure 4D, where only a minor subset of chromosomes are shown). Also, the authors should provide movies stepping through the z-stacks so that the readers can see the chromosomes and count them themselves. This may seem like overkill, but strong evidence is needed if the authors are attempting to dispute a prior published study. (If it turns out that this mutant does not have 24 chromosomes as the current version of the manuscript claims, the conclusions need to be substantially adjusted.) Finally, it is also important to include a better discussion of the Severson paper and provide an explanation for the discrepancies between this study and theirs so the reader can better understand this section of the manuscript.

2. The Materials and Methods section needs more details. For example, the only imaging described is single-plane live imaging, but in some of the figures the authors show sum projections of z-stacks (e.g. Fig 2B, D). Information about the step size and acquisition of these images should be added, as well as details such as how the chromosomes were counted (e.g. Were they counted from the summed images or from going through individual stacks? How did the authors ensure they were counting two independent chromosomes vs. two that are connected? What stage was assessed?).

3. Similar to point #2 above, the Materials and Methods has no information about the generation of the AID strains. Have all of these strains been published before? If so, cite the relevant references in the strain list (Table S1). If not, the authors need to discuss how these strains were made (which terminus were the tags inserted into, what linkers, if any, were used, etc.), and also present a characterization of these strains. Sometimes adding a degron tag and/or GFP can alter protein function. The authors should present brood size and embryonic lethality information, to increase confidence that the tags are not affecting function and impacting the results, as well as western blots to assess the level of depletion they are achieving (especially since some of the phenotypes are not 100% and the authors note that they may not be achieving complete depletion - e.g. page 10 lines 11-12 and page 13 line 13).

4. The authors cite Severson (2009) as showing that spo-11 rec-8 double mutants “retain COH-3/4 on pachytene chromosomes” (page 8 lines 15-18). However, I don’t see that data in the cited Severson paper. If this has been shown somewhere, the proper reference should be used. However, even if it has been shown, is it known whether COH-3/4 remain on chromosomes during spindle assembly (the stage the authors are assessing in Figure 2C)? The authors need to show this (or cite the relevant study) if they want to claim that chromosomes in spo-11 rec-8 have COH-3/4 at the stage they are analyzing.

5. The MEI-1 data in Figure 2E is confusing. The authors claim that the chromosomal pixel intensity divided by the cytoplasmic intensity was not significantly increased, but the signal is so much brighter in the representative image that this is hard to believe. I couldn’t find any information in the Materials and Methods about how this quantification was done - adding this information is essential for the reader to evaluate this data. Also, even if the interpretation of the authors is correct (that the increased contrast of chromosomal MEI-1 is due to a decrease in microtubule-associated MEI-1), couldn’t this result still be real (i.e. couldn’t mistargeting excess MEI-1 to the chromosomes affect the ability of spindles to assemble)? The authors should look at MEI-1 localization in the quadruple mutant that is unable to assemble a bipolar spindle, and in other mutants where cohesin is disrupted and bipolar spindles cannot form (e.g. the bir-1 mutant and haspin(degron)), to see if increased MEI-1 chromosomal staining is correlated with spindle defects.

6. Figure 4: The authors show that in spo-11 rec-8 mutants, some chromosomes have bright AIR-2, and some only have diffuse AIR-2. However, if I understand correctly, all of these chromosomes have “non-cohesive cohesin” that is “sufficient for bipolar spindle formation” (Figure 3). Do the authors have an explanation for why some of these chromosomes are able to pattern AIR-2, while others load diffuse AIR-2 (as they show in Figure 4)? This should be explained somewhere in the text, since it is hard to follow the logic as currently written. Moreover, related to this point, I don’t understand the sentence (page 13 lines 15-19) that states “the subclass of Aurora B that is recruited to chromosomes by cohesin and haspin-dependent phosphorylation of histone H3 is required for bipolar spindle assembly…”. If cohesin recruits Aurora B to chromosomes, then shouldn’t all chromosomes properly load AIR-2 in the spo-11 rec-8 mutant, since the authors showed that SCC-1 is present on all chromosomes? It is possible there is a logical explanation for this that I missed, but I think the authors should explain this better to help the reader follow the logic of the experiments.

7. Figure 4D, 4F: The localization of AIR-2 on chromosomes (diffuse vs. patterned) is not easy to see in the images presented. Also, it would be useful to know, within a given oocyte, how many of the chromosomes have diffuse vs. patterned AIR-2. In the quadruple mutant, it looks like all are diffuse, but in spo-11 rec-8, the image in 4D makes it look like some are patterned and some are not. More oocytes should be assessed for each mutant (only 5-6 oocytes are quantified and pooled in Figure 4C and 4E; more should be assessed) and in addition to the pooled data presented, the authors should report the numbers for each oocyte with appropriate statistics, so it is clear how much this number varies from oocyte to oocyte.

8. In Figure 5, the authors characterize a bir-1 mutant that, based on work in other organisms, should not properly localize to chromosomes (page 12, lines 15-18). However, this has not been shown in C. elegans. The authors should stain for BIR-1 in the mutant, to confirm that there is no chromosomal staining (to make sure that the interpretation of this figure is correct).

9. Figure 6C: In the results section (page 14 line 7-9) it is stated that microtubule bundles in the bir-1 mutant do not coalesce, but the movie shown only goes to 1:30, when control spindles have not coalesced either, so this is not convincing.

10. The different patterns of MCAK on univalents in Figure 7B and 7D are hard to distinguish - even the zoomed images just look like dots or blobs, making it hard to tell what MCAK actually looks like. Higher resolution images are necessary for the reader to understand this localization pattern. Also, similar to point #7 about Figure 4, the authors should increase the number of oocytes analyzed and also report, within a given oocyte, how many chromosomes have patterned vs. diffuse MCAK.

11. The CLS-2 data in Figure 8A is not convincing. Specifically, page 16 lines 10-12 state “cohesin-dependent AIR-2 excludes CLS-2 from the midbivalent ring…”. However, the data presented in Figure 8A does not show that CLS-2 localizes to the ring in the bir-1 mutant - there is only a small amount of CLS-2 on the sides of the bivalent, but this does not extend all the way across, as would be expected for a ring protein. It has been shown that when you deplete AIR-2 to prevent formation of the RC, the midbivalent break between the two homologs is gone (Monen 2007 and Divekar 2021; this is also apparent in the mCH:his images in Figure 8A). Under these conditions kinetochore proteins change from two distinct cups, to one continuous pattern surrounding the bivalent. This is what I think the authors are seeing in Figure 8A. CLS-2 is not loading onto the ring (because there is no ring complex if the CPC is not present); instead, the kinetochore staining just spreads further into this middle region because it coats the entire bivalent. If the authors really want to demonstrate that CLS-2 has RC localization, they would have to show more convincing images, with staining all across the midbivalent region, and also co-localization with another RC component.

12. Related to point #11, the localization of CLS-2 to the midbivalent has not been replicated by others since the Dumont 2010 study, so it is unlikely that CLS-2 is an RC protein (see Pelisch et.al. 2019, Figure 6A, and the control images for Figure 8A in the current manuscript). However, in the discussion CLS-2 is stated to be a RC protein (page 19 lines 19-20) - this should be corrected and Pelisch should be cited.

13. The authors show that there is a positive correlation between the presence of AIR-2 and the recruitment of CLS-2 into larger spheres (Figure 8A). However, is it possible the “larger spheres” just represent a difference in chromosome organization in the presence and absence of AIR-2, rather than a specific effect on patterning CLS-2? Maybe the chromosomes are larger in the presence of AIR-2? The authors should measure chromosome diameter in the presence and absence of AIR-2, to rule out this possibility. Moreover, they should provide a possible explanation for the larger diameter of CLS-2 - this reviewer was left wondering what this pattern meant…what is the significance of a “larger diameter”?

Other points:

- It is hard to keep track of chromosome configurations in the various chromosome structure mutants, especially for people outside the field. I suggest adding more diagrams, similar to those shown in Figure 1A comparing WT and rec-8, but for the other chromosome structure mutants as well (spo-11, spo-11 rec-8, and the quadruple mutant). However, the authors should be careful making these diagrams because in Figure 1A there are thin white lines down the center of some of the single chromatids (e.g. dividing the single chromatids at the end of Anaphase II in the WT and dividing the single chromatids in the rec-8 mutant) – when I first looked at them, I thought that these were put there intentionally to distinguish sister chromatids from each other; it took me awhile to figure out that they were probably mistakes. I suggest remaking this diagram to remove these thin lines.

- Abstract line 13-15: “which regulated the localization of the spindle assembly factors CLASP-2 and kinesin-13 to mediate bipolar spindle assembly.” This makes it sound like these are the key factors that mediate spindle assembly in your mutants, which has not been demonstrated. Moreover, both CLASP-2 and kinesin-13 target to chromosomes lacking AIR-2 (the pattern or circumference is just different), so I am not entirely convinced that the changes in localization observed would affect spindle assembly.

- Figure 1D: It would be helpful for the reader to show ASPM-1 in a control embryo, for comparison.

- Page 10 lines 18-20. This sentence talks about PDS5, but there is no reference, and it is not clear what organism is being discussed - has this been shown in C. elegans? If not, it may be best to remove discussion of PDS5 because it might confuse the reader. There is a similar reference to PDS5 on page 13 lines 14-15.

- Figure 3C: The authors image LIN-5 in Figure 3C and 5D (presumably using it as a pole marker) but they provide no information in the results section about what LIN-5 is or why they are imaging it; this information should be added to make the manuscript more accessible to non-experts.

- Figure 4B: the authors state in the text that AIR-2 is bright on microtubules in Anaphase II (page 11 line 12) but the image shown appears to be end-on and therefore gives the impression that AIR-2 is on chromosomes not microtubules. This localization should be shown more convincingly. Figure 5B has the same problem - AIR-2 appears to overlap with chromosomes in Anaphase II but the text says that AIR-2 is on microtubules.

- Fig S1: I was confused by the discussion of this figure in the results section. Page 12 lines 7-8 states that “sperm-derived paternal DNA within meiotic embryos recruited maternal GFP::AIR-2 but lacked detectable cohesin and did not promote spindle assembly”. However, the authors should explain the experiments presented in Figure S1 in more detail if they want readers to understand this point (the mating experiments are not described except in the Figure S1 legend, and this is not enough information to make this experiment accessible to a broad audience).

- The section on spindle assembly (starting on page 13 line 21) does not reference prior studies describing the wild type spindle assembly pathway - for example, page 14 lines 3-4 notes that microtubule bundles normally arise within the germinal vesicle, but prior studies describing this are not referenced (Wolff et.al. and Gigant et.al. 2017).

- The authors note that MCAK targeting to the midbivalent is dependent on bir-1 (page 15, lines 12-14). Since it has been shown that MCAK targeting to this region requires AIR-2 (Divekar, et.al. 2021), this previous (similar) result should be cited.

- Figure 7: the images of MCAK and AIR-2 colocalization are small and hard to interpret. Does the MCAK pattern match the AIR-2 pattern? Higher resolution images are needed to better understand this result.

- Some of the reported Ns are low. For example, one of the conditions in Figure 3B and 8C only has 3 embryos. A minimum of 5 should be reported (though for some experiments, where there is more variability, this should be even higher; see major points #7 and #10).

Typos:

- Page 6 line 4: “individual” is misspelled

- Page 10, line 5: “AAID” should be “AID”

- Page 12 line 9: “Fig S3” should be “Fig S1”

- Page 15 line 17: “Fig. 6B, C” should be “Fig. 7B, C”

- Page 15 line 20: “Fig. 6D, E” should be “Fig. 7D, E”

- Page 16 line 7: “Fig. 7B, C” should be “Fig. 8B, C”

- Page 16 line 7: “Fig. 7D, E” should be “Fig. 8D, E”

- Sometimes you use “Ran” and sometimes “ran”. Make this consistent.

- Figure 2E: label the top row “merge”

Reviewer #3: The results in this manuscript reveal a role for cohesins in promoting the localization of the chromosome passenger complex (CPC) to the chromosomes and spindle assembly. A key result is that when cohesins are absent, an amorphous cloud of microtubules is observed instead of a bipolar spindle. In MI, this requires depleting both rec-8 and coh-3/4. In MII, because coh-3/4 is normally absent, depletion of only rec-8 leads to a loss of spindle formation. This paper includes a nice result showing auxin-induced depletion of SMC-1 leads to a similar phenotype. In addition, there is strong evidence linking cohesins to CPC recruitment (eg. Fig 4). The main conclusions of this paper are supported by the data. A significant problem, however, is redundancy between figures, lack of raionale for certain genotypes, and a couple experiments that do not show exactly what the authors claim, or the implications are ambiguous. These issues and others listed below should be addressed prior to publication. In short, this paper could be improved with a more concise presentation.

1) Why is the spo-11 mutation in most of the cohesin genotypes. For example, why is there no rec-8; coh-3 coh-4 mutant. It seems that the presence of a spo-11 mutation should not be required to observe the defects in spindle assembly. Besides the presence or absence of the spo-11 mutant may not effect spindly assembly, the results in Fig 2C seem to duplicate results in Fig 1. What does Fig 4F (spo-11 rec-8) show that is not shown in Fig 4B (rec-8) (this results is also not measured or quantified and it should).

2) Figure 1D or E lacks controls. For example, the localization of ASPM-1 is not shown in controls. There is also no rationale given for examining MEL-28 localization. What does this result show? The results in Fig 2E are more relevant and could be move to Fig 1 instead (see below). The conclusion is the same as with ASPM-1, and 2E has a control.

3) Are all the image slices needed in some of the figures, like in Fig 2? I understand this is to see how many chromatids there are, but the key result here is the microtubules, not the number of chromatids. The extra slices could be in a supp figure.

4) Pg 10, line 2: is there evidence that coh-3/4 is a target of separase? If so, this should be cited. In contrast, there are meiotic cohesins that do not appear to be cohesin targets.

5) Some stats in Fig 4C are borderline (p=0.07) although the most important comparison is significant. The sample size is rather small and the authors should try and improve this. In Fig 4D, why is MI not shown?

6) Figure 5: The differences in Fig 5B with AIR-2 localization are a little hard to see. This is a case where side by side images with wild-type would be warranted and quantitation of the AIR-2 localization pattern done if possible. The differences in Fig 5C are measured but seem less impressive and are also prior to meiosis, when different factors could regulate CPC localization.

7) Figure 5D: be clear in the legend why there are two images here. I believe one shows apolar and the other shows multipolar.

8) Figure 6 and text on pg 14: This seems to mostly repeat results in previous Figures. Improve the description to make it clear what is the new finding in this figure, or remove, or move to supp figures.

9) Pg 15, line: Should this be Fig 7B,C? Same for line 20.

10) Figure 7A is important. In contrast, Figure 7B shows cohesin is required for Klp-7 recruitment, not Air-2. For the non-worm person, what is the difference between Figure 7B and D. Both are MI, but one is an embryo. And I really don’t know what the graph in Fig 7E shows. There is no comparison to a control and it is only showing a correlation between Air-2 and Klp-7. But not a relationship to rec-8. I think 7D and E should be deleted.

11) Figure 8B has the same issue as Figure 7B, and 8D,E have the same issues as 7D,E.

12) Pg 16-17: I don’t see how these results exclude a role for the Ran pathway. The fact that the listed SAF proteins are cytoplasmic does not inform on the ran pathway. See below regarding how CPC could derepress cytoplasmic factors.

13) Pg 17, line 20: there is no metaphase III. Use a different term.

14) Pg 18: The authors should cite work from Ohkura in Drosophila (Rome 2019, Beaven 2017) showing strong evidence for suppression of cytoplasmic SAFs that is alleviated by CPC activity. The Xenopus data showing CPC inactivates MCAK is really a isolated result and these experiments did not rule out activation of SAFs. In contrast, the CPC is required for kinetochore assembly is multiple systems.

15) Pg 19 top: Published evidence suggests the ran pathway does not drive spindle assembly in Drosophila oocytes. The authors should discuss how there are several mechanisms to recruit the CPC to the chromosomes in other organisms. The strong role for cohesin described here may be due to the structure of C. elegans meiotic chromosomes. In Drosophila a different chromatin feature rather than cohesins recruits the CPC to the chromosomes (Wang 2021). It is striking, however, that in both C. elegans and Drosophila, recruitment of the CPC is critical for spindle assembly. Mouse also do not require Haspin for spindle assembly.

**Have all data underlying the figures and results presented in the manuscript been provided?**

Reviewer #1: Yes

Reviewer #2: Yes

Reviewer #3: Yes

PLOS authors have the option to publish the peer review history of their article (what does this mean?). If published, this will include your full peer review and any attached files.

Reviewer #1: No

Reviewer #2: No

Reviewer #3: No

---

## [Decision Letter · Decision Letter 1]

2 Jun 2022

Dear Dr McNally,

Thank you very much for submitting your Research Article entitled 'Cohesin is required for meiotic spindle assembly independent of its role in cohesion in C. elegans' to PLOS Genetics.

The manuscript was fully evaluated at the editorial level and by independent peer reviewers. The reviewers appreciated the attention to an important problem, but raised some substantial concerns about the current manuscript. Based on the reviews, we will not be able to accept this version of the manuscript, but we would be willing to review a much-revised version. We cannot, of course, promise publication at that time.

The reviewers all agree that the findings are interesting, but have a significant number of concerns that are not yet addressed by the revision. To move this manuscript forward we ask that you address that reviewers’ comments. Several requests for additional experiments were disregarded in the current revision. Some of these requests have been repeated in the current reviews. The editors agree that these will need to be either experimentally addressed, or very robust reasoning as to why they are not possible will need to be provided (e.g., why fixed sample analysis cannot answer reviewers’ concerns and the authors only consider live imaging analysis? why high-resolution images are not possible using fixed samples?). Other comments are requesting more details, better presentation and change in wording that all can and should be addressed.

If you decide to revise the manuscript for further consideration at PLOS Genetics, please aim to resubmit within the next 60 days, unless it will take extra time to address the concerns of the reviewers, in which case we would appreciate an expected resubmission date by email to plosgenetics@plos.org.

[LINK]

We are sorry that we cannot be more positive about your manuscript at this stage. Please do not hesitate to contact us if you have any concerns or questions.

Yours sincerely,

Sarit Smolikove

Guest Editor

PLOS Genetics

Gregory P. Copenhaver

Editor-in-Chief

PLOS Genetics

Reviewer's Responses to Questions

**Comments to the Authors:**

Reviewer #1: The resubmitted manuscript from the McNally lab has undergone significant revisions, including textual changes in response to reviewer’s comments that greatly improve the readability of the manuscript and the clarity of certain key points; new analyses and/or figures of prior datasets that address some of the reviewer concerns; and an increase in numbers of oocytes analyzed in certain cases where the number was initially quite low. Consequently, the manuscript is much improved. Unfortunately, many important reviewer comments have been inadequately addressed, and some textual changes raise new concerns.

My page numbers below refer to the position in the 128 page long combined PDF, not the page numbers that appear "printed" on each page, which are inconsistent between the section showing the revised draft without changes highlighted and the version with changes highlighted

Major Comments:

1. P. 109, top: “Bipolar spindle assembly 1 occurs in knl-1(AID) knl-3(AID) tir1 worms (Danlasky et al., 2020) and in worms with no degron tag treated with auxin using the same protocol. Thus the spindle assembly defect observed for hasp-1(AID) cannot be due only to non-specifc effects of auxin or the 1% ethanol solvent.” This text was added to give greater detail regarding controls that were done for the auxin degron experiments. It essentially says that experiments done by other people using other lines did not reveal any issues. This does give some level of confidence that the observed results are not a result simply of growth on plates with auxin or the carrier. However, a proper control would be done by the same person, in parallel with the experimental group, using the same batch of media, and would show the phenotype of worms expressing the SAME tagged protein and TIR1 on “normal” plates and worms with TIR1 but without the tag on the same batch of auxin plates. Because proper controls were not done, the word “cannot” is untrue and must be replaced with something else. Perhaps, “Thus, the spindle assembly defect observed for hasp-1(AID) likely does not result from non-specific effects…”

2. Also regarding controls, text was added to the legend for Figure 5 (p. 123) to specify where “control images” are in earlier figures. Unless the experiments were done at the same time, and with only one variable changed from the experimental group, I do not believe the earlier images can accurately be called control images. Moreover, the legend states that “Control for 5D is in Fig. 3C.” Fig. 5D shows mCH::H2B and eGFP::LIN-5, while Fig. 3C shows mKate::TBB-2; SMC-1::AID::GFP; eGFP::LIN-5 in the presence and absence of auxin. These are completely different strains, although I assume the lin-5 transgene is the same. 3C most certainly should not be called a control for 5D. Although it is clunky, the authors must describe the comparisons accurately. Perhaps “eGFP:LIN-5 localization in a different strain, in which a different protein is tagged with AID, is shown in Fig. 3C.” Similar wording is needed to replace “control” for the other parts of this figure to avoid the risk of misleading the reader. A better alternative would be to do the proper controls.

3. Regarding my comment 11 in the first review: The suggestion that the dependence of bipolar spindle assembly on cohesin might prevent the formation of a metaphase III spindle is still quite confusing. In metaphase I and II, cohesin is not required for spindle MT assembly, just for bipolarity. There is never a third round of spindle assembly/MT assembly between anaphase II and pronuclear migration, even in cohesin triple mutants. Why would a requirement for cohesin in promoting bipolarity matter when a third round of MT assembly doesn’t occur? Adding the word "bipolar" or changing the wording from “meiosis III” to something different doesn’t increase the plausibility of this model.

4. Responses to several reviewer comments stated that experiments could not be done because of technical limitations, when the experiments could easily be done if immunofluorescence were used instead of live imaging. These include:

a. Reviewer 2 comment 3 suggested evaluating the level of depletion by AID using a Western blot. The authors response is correct that Western blotting might not give a clean result because TIR1 is expressed only in the germline while the AID-tagged protein is expressed in both soma and germline. However, level of knockdown could be addressed by fixing worms and staining with antibodies to SMC-1 and HASP-1 (SMC-1 antibodies definitely exist; if antibodies to HASP-1 don’t yet exist, there are great antibodies to the Histone H3-PhosT3 mark that HASP-1 makes and to the AID tag used in this study).

b. Reviewer 2 comment 10: patterns of MCAK on univalents are hard to distinguish. It is undoubtedly difficult to get good images of this via live imaging because as the authors state in their response, chromosomes and embryos are moving. However, there are very good antibodies to MCAK, and movement would not be a problem in fixed and stained gonads, so the reason given for not addressing the issue is not convincing.

c. Same for Reviewer 2 comment on Fig. 7 in “Other points.” The concern could easily be addressed by immunofluorescence.

d. Reviewer 1 point 6: do chromatids with intense AIR-2 signal also have high levels of COH-3/4. Although this could not be addressed in live imaging because COH-3::mCherry was too dim, there are very good antibodies for both AIR-2 and COH-3/4, so the reason for not addressing this is not convincing.

Minor Comments:

1. Figure 1B: I agree with reviewer 2 that showing a similar cartoon for spo-11 rec-8 and for spo-11 rec-8 coh-4 coh-3 would be quite helpful for unfamiliar readers to understand the model for these mutants. Also in this figure, the legend does not state what is meant by the blue and green lines. What are “blue cohesin” and “green cohesin”?

2. I agree with the authors that it is important to mention PDS5, since it is the connection between haspin and cohesin. However, I also agree with reviewer 2 that it is important to state in which organisms/cell types the role of PDS5 has been shown and whether it has only been shown in mitosis or also in meiosis. “in mitotically proliferating human cells…”

Reviewer #2: This revised manuscript is much improved, and the major issues have been fixed. Notable improvements include increasing the number of n’s in key figures, better explanations of the quantification schemes, and quantification of the number of chromosomes with patterned Aurora B (Figure S2). I have a few remaining suggestions to improve the manuscript prior to publication, which are all minor compared to the original review and do not require any additional experiments.

Specific points:

- In Major point #5 from my original review, I had noted that it was confusing that the MEI-1 signal looked bright on chromosomes, yet the quantification suggested that it was not increased. I appreciate the addition of details to the materials and methods about how this quantification was done, as well as the explanation provided in the response document explaining the issue that arises when 16 bit images are converted to 8 bit – these clarifications made the data more convincing to me. However, I predict that some future readers will be confused by the images presented, as I was initially, without the provided explanation for how the chromosomes get brighter during the conversion process. Therefore, I think it would be worthwhile to provide the explanation (given in the response document) somewhere in the manuscript as well (materials and methods or figure legend).

- In Figure 3, two panels are labeled “A”. Reviewer 1 commented on it, but this was not fixed. (In my readthrough of the revision, I noticed this too, and I agree with reviewer 1 that it is confusing.)

- Page 10, lines 14-16: the text states that “SMC-1:AID:GFP was found on control metaphase I and metaphase II chromosomes and metaphase I chromosomes of spo-11 rec-8 mutants…” However, when looking at the graph shown in Figure 3B, it looks like a fair number of measured chromosomes in the spo-11 rec-8 mutant in MI fall close to a ratio of 1 (which would indicate no enrichment on chromosomes). I am very convinced that many of the chromosomes have SMC-1, but the raw data suggest that some chromosomes do not. If the authors think this is a valid point (i.e. they agree that some chromosomes may not be enriched for SMC-1 in this condition), then they should add the word “some” or “most” chromosomes when describing their data, and then comment on why some chromosomes may not have SMC-1 (since this is not obvious to this reviewer). But if the authors think that all chromosomes have SMC-1 and there is a different explanation for the chromosomes with a ratio close to 1, then they should provide this explanation in the materials and methods, when describing their quantification scheme.

- Page 10, line 20: there is a reference to smc-1/him-1, but non-worm people will not know what him-1 is. Either explain this or remove the “/him-1”.

- There is inconsistency in how “Aurora B” is written. In the discussion in particular, “auroraB” is used frequently. Unless there is a specific reason for this, I suggest “Aurora B” should be used throughout the manuscript.

- I had suggested that the authors provide more diagrams, similar to the ones in Figure 1A, showing chromosome structure in the other mutants (spo-11 rec-8, and the quadruple mutant). Although the authors did not take my suggestion, I still think that this would help readers who are not C. elegans meiosis experts keep track of chromosome organization in the various conditions, helping them understand the results. Therefore, I would like to again gently encourage the authors to consider this. However, I do not consider this essential for acceptance, so I leave it up to them.

- I still think that it would be helpful if the authors state in the text what organism the PDS5 statements are referring to (in the Yamagishi 2010 reference). Without stating this, readers might think that the sentences are referring to C. elegans PDS5.

Typos:

- Pg. 4 line 19: Ran should be capitalized

- Pg. 16 line 7/8: the word “or” is duplicated

- Pg. 20 line 10: “non-phosphorylatable” is misspelled

- Pg. 24 line 17: “non-specific” is misspelled

- Pg. 37 line 23: “dependingon” should be fixed

Reviewer #3: This is a very interesting paper demonstrating a link between meiotic cohesins, CPC recruitment, and spindle assembly. The authors have made many changes in response to the previous review, but there are still some issues they did not adequately address.

1. The author response that the key comparison is between spo-11 rec-8 and spo-11 rec-8 coh-3 coh-4 does not address the original question of why spo-11 is in there. What if spo-11 influences the phenotype? Just one image of rec-8 coh-3 coh-4 would suffice. It may also help the reader to know why the spo-11 mutation was included. For example, to easily see the effects of the cohesin mutants.

5. Minor point, the reason to show MI images is that the regulation of CPC is known to be different between prophase and metaphase. Additional factors can recruit the CPC in metaphase. And clearly MI is the more relevant time point (see also previous comment #6 and Fig 5C). Agreed the results could easily be the same. Just a thought. From the reader’s perspective, they may wonder, why 4D is one stage while 4F is another (or why 5C is at this stage). The authors do not do a good job explaining this.

Can the authors speculate on why in 4C the chromosomal Air-2 is higher in rec-8 MI relative to controls? Also on Figure 4, can they comment on why the spo-11 rec-8 genotype in 4E show a mix of Air2 intensity, some like the control, some like spoc-11 rec-8 coh3/4.

10) Fig 7A confirms previous results the authors cite. The results in B and C are consistent with these observations but indirect. Therefore, the statement in lines 4-6 pg. 17 applies to Fig 7A only. 7B does not test Air-2 and 7C is only a correlation. However, the author response did not really answer the original question. Fig 7C needs a control. What if the KLP-7 measurement looks the same in wild-type? Similar comment for 7E. And shouldn’t the data in 7B-C also be extractable from 7D? Its also not clear why Klp-7 was not observed in rec-8 coh-3 coh-4, which should have the most dramatic effect.

12) The authors present no evidence on the Ran pathway, even though I agree with the conclusions. For example, being cytoplasmic/ outside the nucleus does not show something is not regulated by the Ran pathway. I think these results are interesting but not because of Ran. Its because these SAFs must be inactive while in the cytoplasm, to avoid bundling of microtubules prior to spindle assembly. And it’s not NEB that triggers spindle assembly, it is exposure of these SAFs to the chromosomes (ands their cohesin recruited CPC) that activates them.

**Have all data underlying the figures and results presented in the manuscript been provided?**

Reviewer #1: Yes

Reviewer #2: Yes

Reviewer #3: Yes

PLOS authors have the option to publish the peer review history of their article (what does this mean?). If published, this will include your full peer review and any attached files.

Reviewer #1: No

Reviewer #2: No

Reviewer #3: No

---

## [Decision Letter · Decision Letter 2]

26 Sep 2022

Dear Dr McNally,

Thank you very much for submitting your Research Article entitled 'Cohesin is required for meiotic spindle assembly independent of its role in cohesion in C. elegans' to PLOS Genetics.

The manuscript was fully evaluated at the editorial level and by independent peer reviewers. The reviewers appreciated the attention to an important topic but identified some concerns that we ask you address in a revised manuscript.

Only one of the reviewers identified places that requires addressing. I believe that these few comments can be easily addressed via text edits, including the clarification question of reviewer 2 [it could be that including the description of what time indicates in M&M is not sufficient to make it clear to the reader and/or statement comparing time by the arrival of the spindle at the cortex (the time on the figures) to the time of ovulation (the time in the text) and how variable it is].

Additional typos are:

Page 8 line 16- Microtubules instead microtublesPage 16 line 20 imaging instead imagngPage 42 line 9 occurred instead occuredpage 44 line 3 expressing instead expresingFig. 2D is called before 2B and 2C, the order of figure 7 panels calls not right

We therefore ask you to modify the manuscript according to the review recommendations. Your revisions should address the specific points made by each reviewer.

[LINK]

Yours sincerely,

Sarit Smolikove

Guest Editor

PLOS Genetics

Gregory P. Copenhaver

Editor-in-Chief

PLOS Genetics

Reviewer's Responses to Questions

**Comments to the Authors:**

Reviewer #1: The revised manuscript includes new data that address the most critical referee comments and suggestions made during the second round of reviews. The manuscript is much stronger and more easily understood because of these revisions, and the data and conclusions regarding cohesin-dependent but cohesion-independent roles for cohesin in spindle assembly are quite convincing and interesting. My concerns have been adequately addressed.

Reviewer #2: The reviewers have addressed my concerns in this revised version. In my final readthrough I caught a few minor errors that I suggest correcting. Otherwise, I am supportive of publication.

- The section describing Figure 2 (starting on line 9) describes events happening in live movies and refers to the timing of events. However, some of the timestamps don’t match when comparing between panels. For example, page 8 line 11 states that microtubules assemble at a similar time after ovulation as control, but if you look at the timestamps in panels 2B and 2D, it does not seem similar (Metaphase I is at -3:20 in panel B, but at 2:00 in panel D). Similarly, the next sentence states that the amorphous cloud shrinks in similar timing, but lists two different timepoints (7:50 and -1:10). Why are these timestamps so different if the authors are claiming they are “similar”? It is possible that this timing difference is explained somewhere, but if so, I missed it. I think it would be useful to add some clarifying information to the figure legend, which is where I think most people would look for this type of information if they are confused.

- Figure S2A does not have a figure call-out in the results section.

- I think that “S5C” on page 14 line 10 is supposed to be S5B. Also, in Figure S5 itself, I think the “B” needs to be moved up (it is placed below labels that should be part of the panel).

Reviewer #3: The authors have done a satisfactory job revising the manuscript. I have no further comments.

**Have all data underlying the figures and results presented in the manuscript been provided?**

Reviewer #1: Yes

Reviewer #2: Yes

Reviewer #3: Yes

PLOS authors have the option to publish the peer review history of their article (what does this mean?). If published, this will include your full peer review and any attached files.

Reviewer #1: No

Reviewer #2: No

Reviewer #3: No

---

## [Editor Report · Decision Letter 3]

10 Oct 2022

Dear Dr McNally,

We are pleased to inform you that your manuscript entitled "Cohesin is required for meiotic spindle assembly independent of its role in cohesion in C. elegans" has been editorially accepted for publication in PLOS Genetics. Congratulations!

In answer to the question you posted in your response letter: please contact PloS Genetic office directly (plosgenetics@plos.org) to enable them to help you with your image file submissions.

Yours sincerely,

Sarit Smolikove

Guest Editor

PLOS Genetics

Gregory P. Copenhaver

Editor-in-Chief

PLOS Genetics

Comments from the reviewers (if applicable):

**Data Deposition**

http://datadryad.org/submit?journalID=pgenetics&manu=PGENETICS-D-22-00290R3

**Press Queries**

---

## [Editor Report · Acceptance letter]

18 Oct 2022

PGENETICS-D-22-00290R3 

Cohesin is required for meiotic spindle assembly independent of its role in cohesion in C. elegans 

Dear Dr McNally, 

We are pleased to inform you that your manuscript entitled "Cohesin is required for meiotic spindle assembly independent of its role in cohesion in C. elegans" has been formally accepted for publication in PLOS Genetics! Your manuscript is now with our production department and you will be notified of the publication date in due course.

With kind regards,

Anita Estes

PLOS Genetics

On behalf of:
